# Probabilistic, high-resolution tsunami predictions in North Cascadia by exploiting sequential design for efficient emulation

Dimitra M. Salmanidou[1], Joakim Beck[2], Peter Pazak[3,4], and Serge Guillas[1]

[1]Department of Statistical Science, University College London, Gower Street London WC1E 6BT, United Kingdom
[2]Computer, Electrical and Mathematical Sciences and Engineering, King Abdullah University of Science & Technology (KAUST), Thuwal, Saudi Arabia
[3]Aon Impact Forecasting – Earthquake Model Development
[4]Earth Science Institute, Slovak Academy of Sciences, Bratislava, Slovakia

**Correspondence:** Dimitra M. Salmanidou (d.salmanidou.12@ucl.ac.uk)

**Abstract.** The potential of a full-margin rupture along the Cascadia subduction zone poses a significant threat over a populous region of North America. Previous probabilistic tsunami hazard assessment studies produced hazard curves based on simulated predictions of tsunami waves, either at low resolution, or at high resolution for a local area or under limited ranges of scenarios, or at a high computational cost to generate hundreds of scenarios at high resolution. We use the GPU-accelerated tsunami
simulator VOLNA-OP2 with a detailed representation of topographic and bathymetric features. We replace the simulator by a Gaussian Process emulator at each output location to overcome the large computational burden. The emulators are statistical approximations of the simulator behaviour. We train the emulators on a set of input-output pairs and use them to generate approximate output values over a six-dimensional scenario parameter space, e.g., uplift/subsidence ratio, maximum uplift, that represent the seabed deformation. We implement an advanced sequential design algorithm for the optimal selection of only
sixty simulations. The low cost of emulation provides for additional flexibility in the shape of the deformation, which we illustrate here, considering two families, buried rupture and splay-faulting, of 2,000 potential scenarios. This approach allows for the first emulation-accelerated computation of probabilistic tsunami hazard in the region of the city of Victoria, British Columbia.

## 1 Introduction

The Cascadia subduction zone is a long subduction zone that expands for more than 1,000 km along the Pacific coast of North America; from Vancouver Island in the North to North California in the South (Fig. 1). The zone lies on the interface of the subducting oceanic plate of Juan de Fuca and the overriding lithospheric plate of North America. Earthquake-induced tsunamis generated from the Cascadia subduction zone pose an imminent threat for the west coasts of the United States and Canada but also other coastal regions in the Pacific Ocean. Historical and geological records show that great plate boundary
earthquakes were responsible for large tsunami events in the past (Clague et al., 2000; Goldfinger et al., 2012). A sequence of great earthquakes has been inferred for the region over the last $\sim 6,500$ years with an average interval rate of $500 - 600$ years (individual intervals may vary from a few hundred to 1,000 years) (Atwater and Hemphill-Haley, 1997; Clague et al.,

2000; Goldfinger et al., 2003, 2012). The most recent megathrust earthquake in the Cascadia subduction zone was the 1700 earthquake, the timing of which was inferred from records of an orphan tsunami in Japan (Satake et al., 1996; Satake, 2003). The moment magnitude (Mw) of the earthquake was estimated close to 9, with a rupture length of ca. 1100 km, likely rupturing the entire zone (Satake, 2003).

There exists a large level of uncertainty with regard to the level of destruction that similar events could cause in the future. Major tsunamis in historical times have not caused significant damage to infrastructure in the west coast of British Columbia (Clague et al., 2003), this is partly attributed to the less densely and scarce populated areas in the region. However, the risk of such an episode nowadays has increased following an increase in urbanisation. The most recent major tsunami impacting the area was generated by the 1964 Great Alaskan earthquake on the 27th March 1964. Although no casualties were reported in Canada, the tsunami caused millions of damage in the west coast of Vancouver Island (Clague et al., 2000, 2003). Studies examining the impact of tsunami in Cascadia have mostly focused on a worst-case scenario potential (Cherniawsky et al., 2007; Witter et al., 2013; Fine et al., 2018); a few probabilistic studies exist, primarily assessing hazard potential on the U.S. coastline (Gonzalez et al., 2009; Park et al., 2017) for a limited number of scenarios at high resolution, or at individual local points (Guillas et al., 2018) for a large number of scenarios but at a moderate resolution of 100 m. Davies et al. (2018) performed a probabilistic tsunami hazard assessment at a global scale.

Probabilistic approaches allow for the exploration of large scenario distributions that benefit risk-informed decision making (Volpe et al., 2019). The probabilistic approach is to treat the uncertain scenario parameters as random variables and then propagate the parameter uncertainty to model the outputs. Uncertainty quantification aims to efficiently estimate the resulting variability in the simulation output, for instance in the simulated maximum tsunami wave heights on a set of locations. Thus, one needs to run the tsunami simulator for many scenarios with parameter values drawn from a chosen probability distribution, defining our prior belief about different scenarios' probability. High accuracy, high resolution computations are especially useful in tsunami modelling studies to assess inundation, damage to infrastructure and asset losses, but also for evacuation modelling. The parameter space dimension is also typically high, and the number of expensive numerical simulations needed to resolve the statistics about the output tends to be large (in the order of thousands for a well approximated distribution, (Salmanidou et al. (2017); Gopinathan et al. (2021)) and hard to materialise as it depends on the available resources, code architecture and other factors. Statistical emulators (also known as statistical surrogate models) can be called to address these issues (Sarri et al., 2012; Behrens and Dias, 2015). We propose using a statistical surrogate approach, also called emulation, in which one approximates simulation outputs of interest as a function of the scenario parameter space. Such approaches have been implemented for uncertainty quantification of tsunami hazard at various settings (Sraj et al., 2014; Salmanidou et al., 2017, 2019; Guillas et al., 2018; Denamiel et al., 2019; Snelling et al., 2020; Gopinathan et al., 2021; Giles et al., 2021).

Statistical emulators are stochastic approximations of the deterministic response. They are used to predict the expected outputs of the response at untried inputs that fall within the prescribed input parameter intervals. Training data, which are the observations of the response at various settings, are used to build the emulators and are thus of paramount importance. In tsunami hazard, where observations of past events are limited, these training data originate from numerical experiments that have been mainly constrained by some physical understanding of the widest range of possible scenarios in order to cover any

possible event through the emulation process, since interpolation, not extrapolation, is the core technique. Extrapolation means predicting outcomes for parameter values beyond the parameter domain on which emulators are designed to interpolate. Since the points representing seabed deformation scenarios are in a bounded parameter domain, emulators can mitigate undesired extrapolation if built on a training design set with good coverage of the domain, particularly if the envelope of the design set is close to the domain boundary. For small design sets, which we consider in this work, sequential design strategies are advantageous as they update the design set to improved coverage, among other desired design features, by conditioning on the current design point locations. The role of experimental design in the scientific studies becomes, thus, critical as it aims to select the optimal sets of variables that contribute to the variance in the response and, in parallel, minimise the numbers of the runs needed for a desired accuracy. Several methods exist in the literature, from which two commonly occurring designs are the fixed/one-shot and the adaptive/sequential design. In fixed designs, such as the Latin Hypercube sampling (LHS), the sample size of the experiments is prescribed. These designs have good space-filling properties, but may waste computational resources over unnecessary regions of the input space. On the other hand, sequential designs adaptively select the next set of experiments to optimise the training data for fitting the emulators. Such a design can be determined by the efficient Mutual Information for Computer Experiments (MICE) algorithm (Beck and Guillas, 2016) that we utilise in this study for probabilistic tsunami hazard prediction in North Cascadia.

Our study builds a methodology that employs existent methods and tools for the design of computer experiments and statistical emulation in order to quantify the uncertainty of tsunami hazard in British Columbia. The objective is to build Multi-Output Gaussian Process (MOGP)[1] emulators and use them for probabilistic, high-resolution tsunami hazard prediction. Other surrogate model techniques have been applied for tsunami or tsunami-like applications, such as polynomial regression (see, e.g., Kotani et al. (2020)) and artificial neural networks (see, e.g., Yao et al. (2021)). For example, Yao et al. (2021) predicted tsunami-like wave run-up over fringing reefs using a neural network approach for approximating the relationship between inputs, including incident wave height and four reef features, and a wave run-up output on the back-reef beach. The authors emphasized that a disadvantage of artificial neural networks is that they are not suitable for small datasets. Owen et al. (2017) demonstrated, by examples involving computer-intensive simulation models, that GP emulation can approximate outputs of nonlinear behaviour with higher accuracy than polynomial regression when considering small- to moderate-sized, space-filling designs.

The benefit of this approach is the use of a sequential design algorithm in the training to maximize the computational information gain over the multidimensional input space and adaptively select the succeeding set of experiments. The vertical seabed displacement over the Cascadia subduction zone was defined by its duration and a set of shape parameters. We develop a site-specific idealised model for the time-dependent crustal deformation along the subduction zone, controlled by a set of shape parameters. In our study, the shape parameters are the model input, and the values define a specific scenario. The tsunami hazard was modelled using the GPU-accelerated nonlinear shallow water equation solver VOLNA-OP2 (Reguly et al., 2018). The acceleration with GPUs makes it computationally feasible to run tsunami simulations on highly refined meshes for many scenarios. By a scenario, here, we mean a specific seabed deformation causing a tsunami outcome. For each location in a

---

[1]https://github.com/alan-turing-institute/mogp_emulator

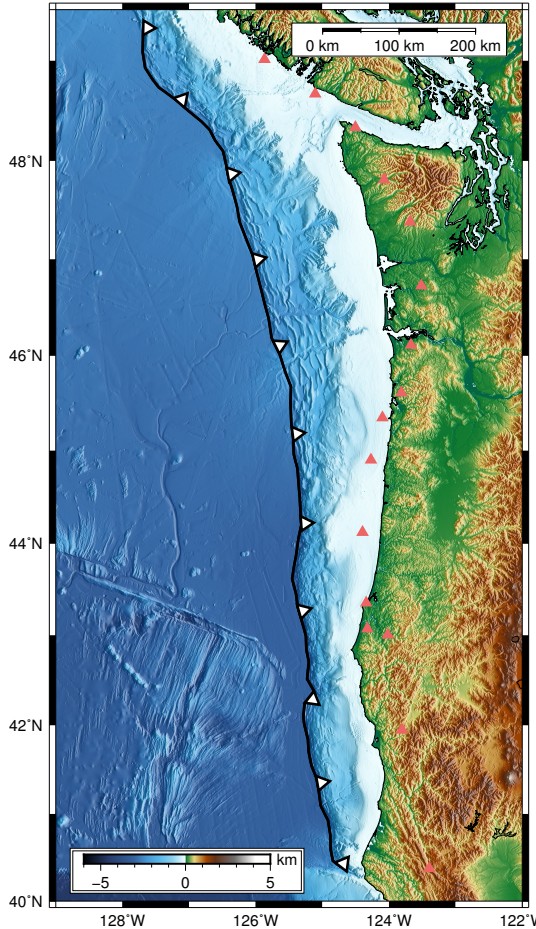

**Figure 1.** The domain of interest. The black line and white arrows depict the location of the trench, the orange triangles show the points used to drive the maximum subsidence. The reference point of the scale bar is assumed to be the bottom left corner of the map.

refined area of 5,148 mesh locations at the shoreline of Southeast Vancouver Island, we create a corresponding emulator of the expensive high-resolution tsunami simulator. The implementation of MOGP emulators finally allows us to predict the maximum tsunami wave heights/flow depths at shoreline level ($H_{max}$) at a high resolution, which can then be utilized to

95 assess the probabilistic tsunami hazard for the region. We note that we compute the flow depth, as opposed to the wave height, for shoreline locations that have elevation above zero. The advantage of the design is that only a relatively small number of expensive tsunami simulator runs, that constitute the training data for the emulators, needs to be performed. A fast evaluation of these emulators for untried input data is then performed to produce approximates of what the tsunami simulator output would have been. The emulators' technique of choice is a Gaussian process regression, which is also widely known as Kriging.

The novelty here is the use of the sequential design MICE by Beck and Guillas (2016) for the construction of the GP emulators of the tsunami model. This is done for the first time towards a realistic case using High Performance Computing (HPC).

A one-shot random sampling for the training (as for example in Salmanidou et al. (2017), Gopinathan et al. (2021) and Giles et al. (2021)) lacks the information gain achieved by sequential design. Concretely, sequential design can reduce by 50% the computational cost, as demonstrated in Beck and Guillas (2016) for a set of toy problems, so applying this novel approach towards a realistic case is showcasing real benefits in the case of high resolution with a complex parametrization of the source. This work differentiates from the previous work by Guillas et al. (2018) on several aspects such as the high-resolution modelling, the sequential design approach, the complexity of the source and the use of the emulators for studying the probabilistic tsunami hazard in the region. The focus of our work is on the methodological aspect of building the emulators and using them for multi-output tsunami hazard predictions. For a comprehensive tsunami hazard assessment realistic modelling of Cascadia subduction interface magnitude-frequency relationships and seabed deformation parametrization needs to be incorporated. The study workflow followed in this study can be divided in three stages (Fig. 2): I) the experimental set-up, II) the experimental design, III) the emulation and its use in hazard assessment. Each stage is described in detail in the following sections.

## 2 Set-up of experiments

To proceed with the numerical experiments, some choices with regard to the input data need to be made a priori. Hence, the first stage of the study deals with the set-up of the numerical experiments. This stage can prove critical as changes in the set-up at a later stage of the study (e.g. emulation stage) might result in the re-initiation of the process. Several aspects need to be considered some of which are: the choice of models and functions to perform the analysis, the input parameters to describe the seabed deformation and their ranges, the data acquisition and processing, and the grid configuration required to best represent the issues in question (see also yellow panel in Figure 2).

### 2.1 Input parameters

Most of the megathrust earthquake scenarios examine the possibility of buried rupture or splay-faulting rupture in the northern part of the segment (Priest et al., 2010; Witter et al., 2013; Fine et al., 2018). Trench breaching rupture scenarios were also studied by Gao et al. (2018). Various studies have dealt with the seabed displacement leading to tsunami excitation in the Cascadia subduction zone (Satake, 2003; Wang et al., 2003). In this study, an idealised geometry in the form of a cosine interpolation is employed for the smooth representation of the seabed deformation along with the deformation from its highest point towards the coast direction. The maximum and minimum points of the interpolation function correspond to the locations of maximum uplift and subsidence respectively (Fig. 1). The points along the trench (60 locations) were defined by the morphological change between the undeformed ocean floor of the subducting Juan de Fuca plate, and the irregular deformed slope of the overriding North American plate. A full-length rupture is computed for all the scenarios presented in the study. The seabed deformations change over time by multiplying an amplitude factor that increases linearly from the initial time set at 0 to the duration value $t$. Seven input parameters were considered to describe the deformation: the total rupture duration, the horizontal distance from the trench to a) the point of the maximum uplift ($Dh_{max}$), b) the point of maximum subsidence ($Dh_{min}$) and c) the point where the deformation stops ($Dd$), the maximum uplift on the trench line ($h_t$), the maximum uplift

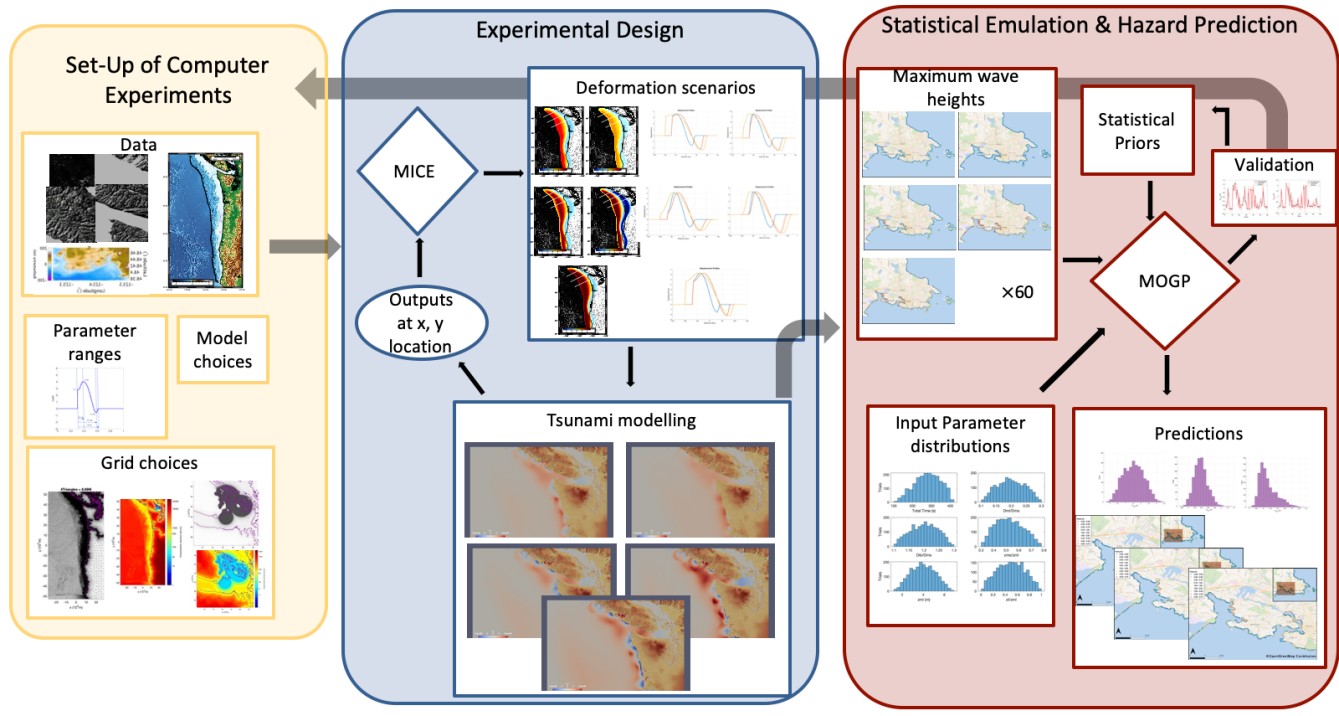

**Figure 2.** The graph of the workflow divides the study in three principal stages that are interlinked: stage 01 (yellow panel) comprises the study specification and set-up of the experiments, stage 02 (blue panel) comprises the study design and conduction of the numerical experiments and stage 03 (red panel) comprises the building of the emulators and their use for prediction. The maps in the predictions section of stage 03 are produced with the QGIS software using as base-maps the Wikimedia[a] layers with data provided by OpenStreetMap contributors, 2021. Distributed under a Creative Commons BY-SA License[b].

---

[a]maps.wikimedia.org

[b]https://www.openstreetmap.org/copyright

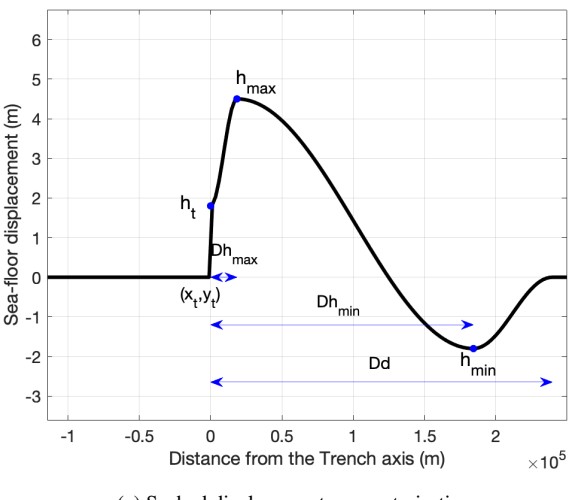
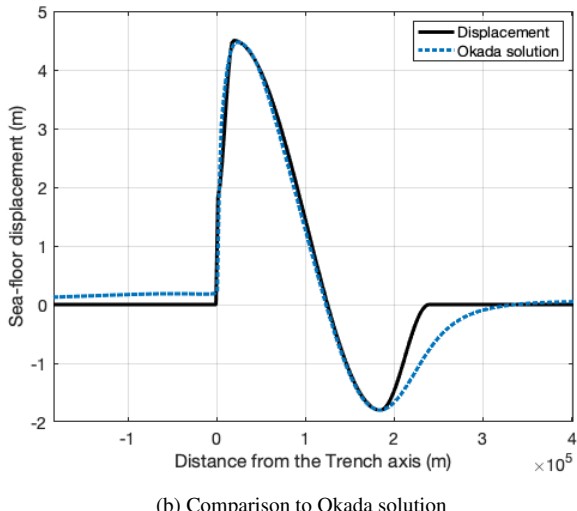

(a) Seabed displacement parameterisation

(b) Comparison to Okada solution

**Figure 3.** Geometry of the vertical seabed displacement (cross-section). (a) Example using $Dh_{max}/Dh_{min} = 0.1$, $Dd/Dh_{min} = 1.3$, $h_{min}/h_{max} = 0.4$, $h_{max} = 4.5$ m and $h_t/h_{max} = 0.4$, (b) comparison to an Okada solution for slip occuring on a fault with geometry similar to the Cascadia subduction interface.

recorded ($h_{max}$) and the maximum subsidence ($h_{min}$) (Fig. 3). The duration, $t$, the maximum vertical displacement, $h_{max}$,

and the ratios of $h_{min}/h_{max}$, $h_t/h_{max}$, $Dh_{max}/Dh_{min}$ and $Dd/Dh_{min}$ are then utilised to model the deformation.

     The choices for constraining some of these variables were motivated by the existing literature. For example, the duration of mega-thrust earthquakes may increase with increasing earthquake size. The co-seismic crustal deformation of earthquakes larger than Mw 8 can usually last for more than one minute (McCaffrey, 2011). The rupture duration recorded during the 2004 Sumatra-Andaman earthquake was approximately 500 s, during which the 1,300 km zone ruptured at speeds of 2.8 km/s (Ishii

et al., 2005). The 2011 Tohoku earthquake, on the other hand, had a rupture duration that might have lasted approximately 150 s (Lay, 2018). These large variations in the rupture duration are not solely dependent on magnitude but on more complex rupture characteristics (Bilek and Lay, 2018; Lay, 2018). Assuming rupture speeds of 2.8-4 km/s, a rupture of 1,100 km in the Cascadia Subduction Zone would yield 275-393 s. A larger time range, $t$, varying between 100 and 420 s is considered for the simulations.

The seabed deformation of a future event in the Cascadia Subduction Zone cannot be predicted with certainty. The 1700 earthquake possibly caused a vertical displacement of ca. 4 m when considering a full length rupture (Satake, 2003). The subsidence of the event was inferred from microfossil data to have ranged between 0.5-1.5 m at several coastal sites in the Pacific Northwest (references in Satake et al., 2003). Similar or larger values have been observed in other great subduction zone earthquakes (Fujiwara et al., 2011; Maksymowicz et al., 2017). Turbidite event history for the Cascadia subduction

interface (Goldfinger et al., 2012) shows that for larger magnitudes the zone predominantly ruptures as a whole rather than concentrating high slip on smaller rupture surfaces. The, recently published, 6th generation seismic hazard model for Canada

by the Geological Survey of Canada includes only full margin ruptures for the range of magnitudes 8.4-9.2 that are important for seismic and also for tsunami hazard assessment. Therefore, for the purpose of our modelling only full margin ruptures are employed and variations of the seabed deformation for such ruptures is considered - the variations are due to different final slip distribution on the full-margin rupture surface. Comparing the typical shape of the displacement against the deformation generated using the Okada solution for a dipping thrust fault, gives a good agreement for the shape of the uplift, albeit a more extensive subsidence (Fig. 3b), the impact of which needs to be further assessed in future research. Leonard et al. (2004) showed that the largest subsidence (0-1 m) concentrated on the west part of Vancouver island, during the 1700 earthquake.

It is estimated, that splay-faulting rupture in the Northern part of the zone could result in an enhanced vertical displacement in the vicinity of the deformation front (Priest et al., 2010). Witter et al. (2013) modelled various deformation models for tsunami excitation with respect to a 1,000 km rupture length. They found that for events with recurrence rates of 425-525 years, splay-faulting may increase the maximum vertical displacement in the northern part of the zone (Olympic peninsula) to 7-8 m with maximum subsidence between 1.5-2.5 m. The maximum uplift and subsidence decrease to ca. 4 m and 1.4 m when moving southward (Cape Blanco) and splay-related displacement ceases below 42.8° (Witter et al., 2013). Based on the above considerations the range of $h_{max}$ was chosen to be 1-8 m, with the ratio of $h_{min}/h_{max}$ estimated to range between 0.3-0.8. A full margin range was specified for the ratio of $h_t/h_{max}$: [0.0, 1.0]. Finally for the distance lengths the ratios of $Dh_{max}/Dh_{min}$ and $Dd/Dh_{min}$ were varied between [0.1,0.3] and [1.1,1.3] respectively. Albeit the larger ranges to train the emulators, information on the source can be interpreted more efficiently in the prediction of the process (Section 4).

## 2.2  Model choices

For the tsunami simulations the numerical code VOLNA-OP2 was used (Reguly et al., 2018; Giles et al., 2020). The code has been employed at several occasions for the numerical simulation and the uncertainty quantification of tsunami hazard (Sarri et al., 2012; Salmanidou et al., 2017; Guillas et al., 2018; Gopinathan et al., 2021). By integrating the bathymetry displacement in VOLNA-OP2, the full tsunami process can be modelled from generation to onshore inundation. The code solves the depth-averaged Nonlinear Shallow Water Equations (NSWE) using a cell-centered finite volume scheme for the spatial discretisation. The 2nd order Runge-Kutta scheme in conjunction with a Strong Stability-Preserving (SSP) method is used for the temporal discretization. The utilisation of unstructured, triangular meshes allows for the incorporation of complex topographic and bathymetric features and accommodates the accurate representation of the region of interest. The VOLNA-OP2 has been massively parallelised and accelerated on general-purpose Graphics Processing Units (GPUs) and has been validated against known tsunami benchmarks and tested for its accuracy and computational efficiency (Giles et al., 2020).

For the tsunami hazard predictions the Multi-Output Gaussian Process (MOGP) emulators are utilised. To perform the analysis we use the MOGP emulation code which is maintained and freely distributed by the Research Engineering Group at the Alan Turing Institute. A Gaussian Process (GP) regression is the core component of the method. The GP fits a specified set of input and output variables using a multivariate Gaussian distribution with given mean and covariance functions, it also allows for the prior choice of the hyperparameters. The hyperparameters are the parameters in the covariance function of the GP regression model. Their values are generally uncertain and can be assigned using priors or fitted to the training data by

maximum likelihood estimation. The benefits of the MO approach is that this process can be run in parallel so that multiple emulators are fitted to the training input and output variables simultaneously while maintaining their independence in the solution. In this study we use as input variables the time and the shape parameters of the deformation (Section: 2.1) and as outputs the $H_{max}$ values observed at the coastline to build the emulators at 5,148 locations.

An active learning approach is employed to sequentially design the training data for the Gaussian process regression. A common approach is Active-Learning MacKay (ALM) where one chooses the design input in each sequential selection that produces the longest predictive variance. In this work, we use the active learning algorithm MICE that provides an informative design of training data for prediction at a lower computational cost than ALM (Beck and Guillas, 2016). The MICE algorithm extended the algorithm for near-optimal sensor placement by Krause et al. (2008), that uses a mutual information-based design criterion, to the setting of design of computer experiments with Gaussian process emulation.

## 2.3 Data and Grid configuration

The digital bathymetry and topography data originate from a compilation of sources that vary in resolution from fine to coarse layers. For the coarse digital elevation layer the GEBCO_2019 Grid product, from the General Bathymetric Chart of the Oceans (GEBCO), is used which has a spatial resolution of 15 arc-seconds. For the high-resolution layer Digital elevation models from two data sets are merged: the Shuttle Radar Topography Mission (SRTM) and the National Oceanic and Atmospheric Administration (NOAA). The SRTM provide topographic data at a spatial resolution of 1 arc-second ($\sim 30$ m). The NOAA DEMs are distributed by the National Centers for Environmental Information (NCEI) and have a spatial resolution that can be as fine as 1/3 arc-seconds ($\sim 9$ m), albeit without a full coverage. The data were interpolated and merged to generate a computational grid with a fine spatial resolution of 30 m in the region of interest that gradually decreases with increasing distance sustaining, however, a 500 m coastline resolution all across the computational domain. The interpolation and mesh generation algorithms were first implemented by Gopinathan et al. (2021) for the numerical simulations and the statistical emulation of tsunami hazard in the Makran subduction zone. The algorithms create a triangular grid by making use of the Gmsh mesh generator[2]. Several design strategies were explored to perform the high-resolution analysis for the coastlines of Vancouver island in a way that satisfies the optimal trade-off between the different mesh set-ups and the affordable size of each run. The domain was, thus, split into 3 sub-domains that focused the high resolution outputs either at the coastlines of SW Vancouver island, or the West or the East part of the Juan de Fuca Strait. The results presented here show the tsunami propagation and inundation in the East part of the Strait; the computational mesh has 8,693,871 elements.

## 3 Experimental Design

To automate the experimental process a workflow was developed exploiting the HPC capabilities. In the core of the workflow (see also blue panel in Figure 2) lies the sequential design algorithm MICE which controls the scenario selection. The numerical experiments are divided in batches with each batch containing a set of five experiments. In the beginning of the process, MICE

---

[2]https://gmsh.info/

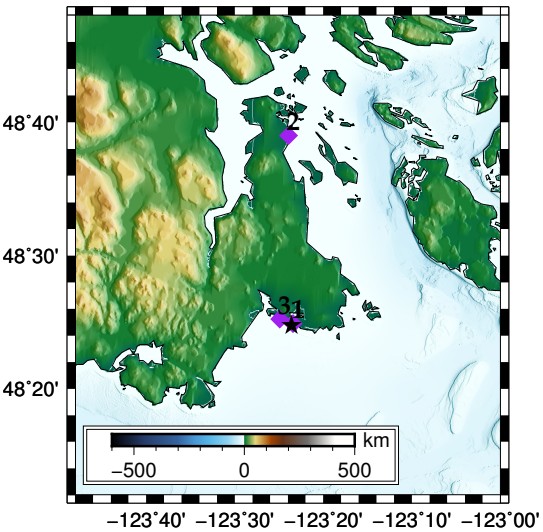

**Figure 4.** The location that drives the design algorithm is indicated with a black star (lon:-123.3934, lat:48.4127). The locations of three emulators (1- lon:-123.3904, lat:48.4149; 2- lon:-123.3989; lat: 48.6657; 3- lon:-123.415478, lat:48.420584) at the cell centres of the grid are displayed with purple diamonds.

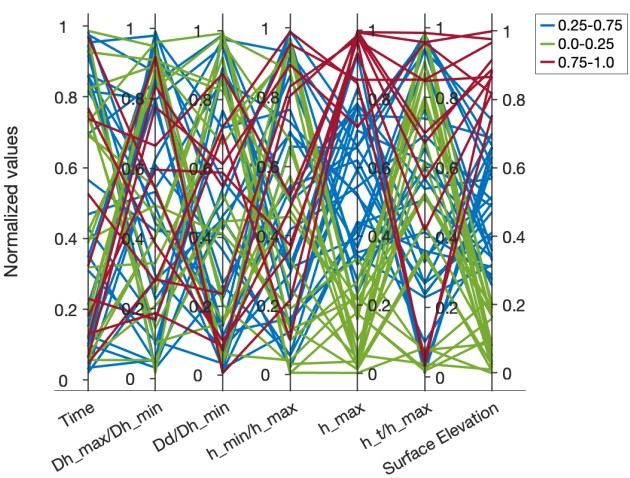

**Figure 5.** The Parallel Coordinates plot shows the choice of input parameters and the output maximum surface elevation at the design location for the 60 experiments. All the data values are normalized.

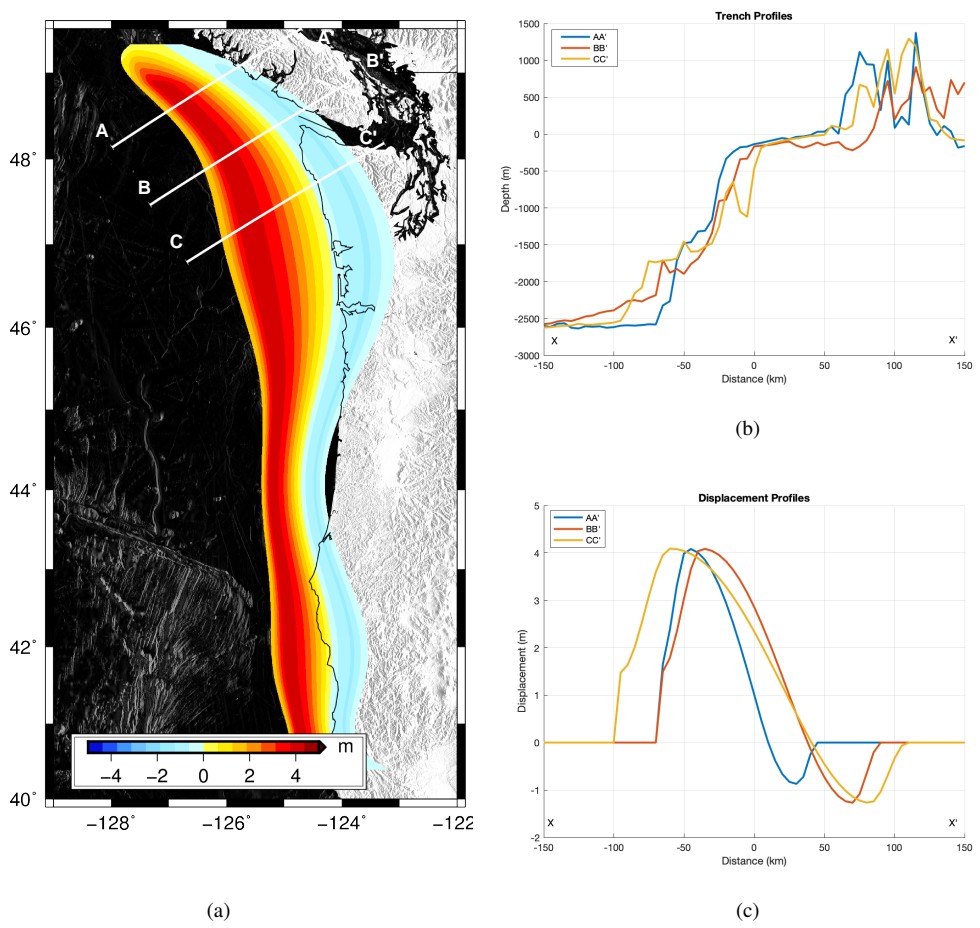

(a)

(b)

(c)

**Figure 6.** Top view of the seabed deformation for trial no. 24 and profiles of the trench and vertical displacement

selects randomly the first set of experiments. Based on the selection of the initial input values (first five runs), the deformation for each scenario is computed. The input files required for the tsunami simulations are then produced, following the selected deformation. In the end of the tsunami simulations, the maximum elevation recorded at one location informative for the design (here we choose lon:-123.3934 and lat:48.4127, see Figure 4) is extracted and used as quantity of interest for MICE (Fig. 2). The algorithm then drives the selection of the subsequent sets in a way that allows for the optimal exploration of the input parameter space and the process is repeated. These iterations occur 12 times for a total of 60 scenarios. Each simulation runs for three hours in real time with a timestep of $dt = 0.01$ s.

The tsunami generation follows the pattern of the bathymetry deformation across the 60 scenarios (Fig. 5). The parallel co-ordinates plot in Figure 5 represents the values of the input parameters selected by MICE and their associated output maximum sea surface elevation as recorded at the design location (black star in Fig. 4). This location drives the experimental design and it was selected as it provides variance in the response driven by each scenario and it is close to the centre of the region of impact. As it directs the sequential design, there is some sensitivity of the design to this point, but not that large, in our opinion, as another point in the region would yield similar results since the influence of the parameters on impact points does not vary significantly. Furthermore, small variations in the design of experiments obtained have little influence on the construction of the emulator, but an agnostic one-shot design (such as LHS where zero design locations are used) would greatly differ from any of the sequential designs obtained by our approach and be less efficient as it would ignore completely the concrete influence of the inputs on outputs to efficiently design the computer experiment. Evidently, adding more locations could improve the design further, but some methodological statistical developments should be first established to decide on a strategy to actually benefit from using more points. All the values are normalised, whereas the surface elevation values are categorised in three bins ([0-0.25], [0.25-0.75], [0.75-1]) for clarity. From the plot, some inference can be made on the influence of the input parameters on the output values. For example, the most influential parameter is the maximum uplift, as higher values result in higher surface elevation (Fig. 5). As an output from VOLNA-OP2 we extract $H_{max}$, as recorded on the cell-centres of the grid all across the domain's coastline. To build the statistical emulators, the $Hmax$ values at the coast have been extracted from each simulation for the geographical region with longitudes of: [-123.49, -123.22] and latitudes: [48.38,48.55]. Within these boundaries, the cell-centres that correspond to the coastline that had recorded an elevation greater than $10^{-5}$ m, for at least one scenario, were considered resulting in 5,148 locations (and subsequently emulators).

## 3.1 A large scenario for initial validation

Looking at a sample scenario demonstrates part of the process at an individual level. Scenario 24 is selected as it is the first scenario in our list of scenarios that has a maximum deformation of ca. 4 m, similar to the maximum uplift in numerical studies of the event (Fig. 6) (Satake, 2003; Cherniawsky et al., 2007). The input parameter values of the scenario are: time $= 281$ s, $Dh_{max}/Dh_{min} = 0.18439$, $Dd/Dh_{min} = 1.18176$, $h_{min}/h_{max} = 0.31049$, $h_{max} = 4.09125$ m and $h_t/h_{max} = 0.36047$. The maximum uplift was used as a guideline for this comparison due to its significant contribution to the tsunami excitation. As the experimental setting is controlled by MICE, the rest of the parameter values of scenario 24 do not necessarily match with the values of other numerical studies. For example, the maximum subsidence of scenario 24 is selected to be around 1.27

| Study | Uplift (m) | Subsidence (m) | Approximate arrival time of Wave Trough (minutes) | Approximate arrival time of Wave Crest (minutes) | Approximate Wave Trough (m) | Approximate Wave Crest (m) |
|---|---|---|---|---|---|---|
| Scenario 24 | 4.09 | 1.27 | 50 | 100 | 0.2 | 1.7-1.8 |
| AECOM, 2013 | 6.2 | 2.3 | Not found | 96 | 1-1.05 | 2.4-2.5 |
| Cherniawsky et al. (2007) | Not found, Mw 9 | Not found, Mw 9 | 50 | 90 | 0.5-0.6 | 1.9-2.2 |
| Fine et al. (2018) | 4 | 2-2.5 | 52 | 88 | 0.96 | 1.6-1.7 |

**Table 1.** Modelling studies of tsunami at the mouth of Victoria Harbor.

m as opposed to 2 m in the buried rupture model by Fine et al. (2018). This causes some discrepancies in the wave signal, the degree of which is not calculated since the scope of this comparison is to do an initial validation of our modelling as opposed to a reproduction of the currently existing work.

The tsunami generation and propagation is shown in the snapshots of Figure 7. A tsunami trough is generated in the area of maximum subsidence and propagates east in the Strait of Juan de Fuca followed by the tsunami crest generated in the region of maximum uplift (Fig.7). The tsunami crest arrives in the Strait approximately 30 minutes after the tsunami generation and has reached the San Juan islands after 110 minutes of propagation (Fig.7). The tsunami trough arrives at the location of the offshore design gauge, near Victoria's breakwater (Fig.8), after ca. 50 minutes of propagation and records -0.2 m. The first wave crest in the gauge is recorded at ca. 100 min at an elevation of ca. 1.8 m (Fig.8). The arrival times come in close agreement with the arrival times computed by Fine et al. (2018) for two rupture scenarios with maximum uplifts of 4 m (buried rupture) and 8 m (splay-faulting rupture). The authors computed arrival times of 52 and 88 min for the first wave trough and wave crest respectively at a similar location (Victoria's breakwater), the corresponding minimum water levels were -0.96 m and 1.63 m (Fine et al., 2018). The maximum wave elevation is in close agreement with the maximum wave amplitude of scenario 24. The discrepancies in the negative wave troughs can be attributed to the discrepancies in maximum subsidence between the two rupture scenarios.

Similar values have been recorded in other numerical studies. Clague et al. (2000) estimated wave run-ups ranging between 1-5 m in the city of Victoria. Cherniawsky et al. (2007) computed a maximum wave elevation of ca. 2 m in Victoria's harbor, with larger values concentrated in small bays around the area. Larger water elevations have been computed by AECOM (2013) for a Mw 9 earthquake, having a maximum uplift of 6.2 m and maximum subsidence of -2.3 m. The earthquake-induced tsunami resulted in maximum water surface elevation between 2.4 and 2.6 m in the harbor openings of Victoria and Esquimalt which increased up to 4.3 m due to resonance in shallow, narrow regions; the computed minimum water levels varied between -1 and -2 m (AECOM, 2013). Figure 8 shows the contours of the maximum elevation of the event, as computed around the area

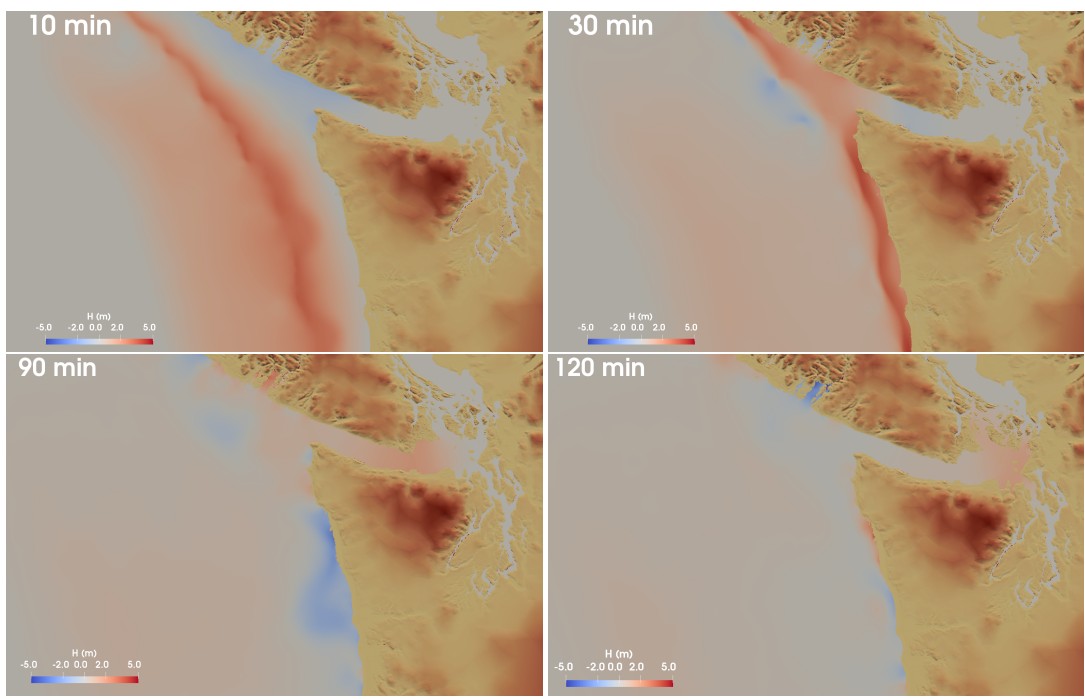

**Figure 7.** Snapshots of the tsunami propagation of scenario 24 at time intervals T=10, 30, 90 and 120 minutes.

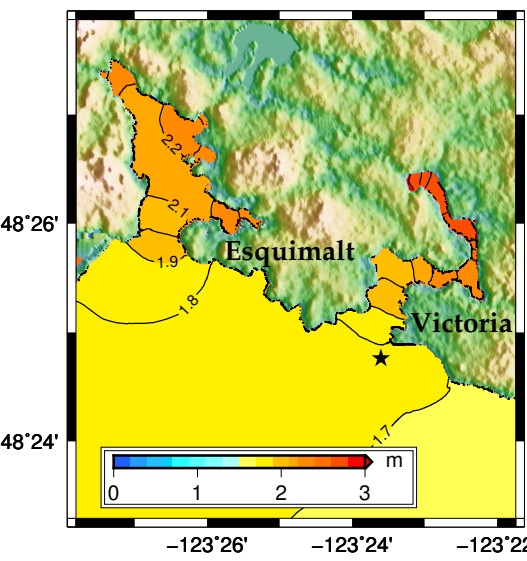

**Figure 8.** The contours of the maximum elevation from scenario 24. The black star denotes the design location used to drive the experiments.

of Victoria and Esquimalt. In the geographical opening of the harbors the maximum tsunami elevation recorded for scenario 24 ranges between 1.8-2 m. However, these values tend to increase inside the harbors which is especially evident in narrow bays and coves. Similar outputs are also observed in the predictions and are discussed in more detail in Section 4.2.

## 4 Probabilistic Tsunami Hazard

To generate the probabilistic outputs the emulators must be built and used. Hence, this stage can be split in three main parts: a) the fitting of the MOGP emulators, b) their utilisation for tsunami hazard predictions at the cell-centres of the grid (see also red panel in Figure 2) and c) probabilistic tsunami hazard calculation via association of the output seabed deformations to earthquake events and their magnitude-frequency relationships so that annual frequencies can be assigned to the calculated inundation depths.

### 4.1 Fitting

The fitting process involves the construction of the emulators using the training data with certain choices on the statistical model. The training data are the input deformation parameters and the numerical outputs of the VOLNA-OP2, represented as $H_{max}$ at the cell-centres of the grid at the coastline, from the 60 numerical scenarios selected by MICE. These are used in conjunction with the statistical choices for the mean and the covariance function. A zero mean function was used for each emulator. The emulators were built employing a squared exponential covariance function with the hyperparameter values estimated from the training data by a Maximum Likelihood Estimation, thus no prior distributions were considered on the hyperparameter values.

The design and the built emulators were validated using the Leave-One-Out (LOO) diagnostics. Following this approach, we build an emulator by excluding each time one simulation from the training inputs and outputs; we then predict the expected outputs for the selected scenario. We test the results at three locations to illustrate the variations in the outputs: one close to the gauge that was utilised in the design, one further away from the design point (location 2) but with elevation similar to the one of location 1, and one (location 3) close to the design location but with a very different elevation (locations 1, 2 and 3 in Figure 4). The plots in Figure 9a, b & c represent the comparison between the numerical response from VOLNA-OP2 (characterised as the "true" response in the graphs) and the predicted response with the variance in these locations. As the plots demonstrate, in some cases the emulator underpredicts the response but there is an overall good agreement between the predictions and the response as the majority of the points is captured by the variance around the predictions (Fig. 9a, b & c). As the waves propagate on land, the prediction becomes more challenging due to even the slightest variations caused by the surrounding topography. The sensitivity of the locations to the variance in the scenarios also plays a significant role. Location 2, for example, does not show large sensitivity to the variation of the parameters, the maximum elevation is close to zero in all of the cases. Location 3 is closer to the source and is experiencing the highest wave run-up and it is, therefore, less affected by these slight variations in the topography but more sensitive to the varying scenarios.

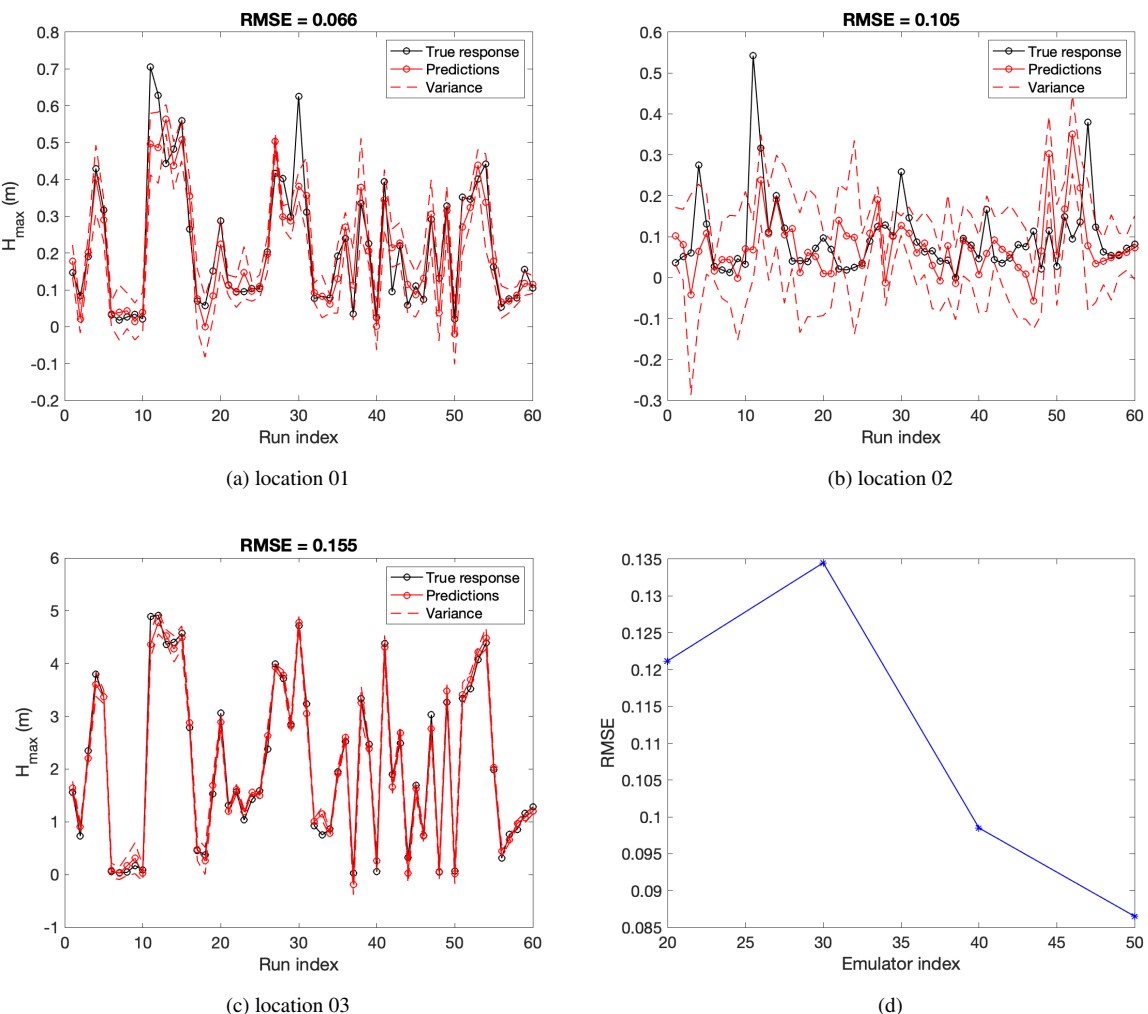

**Figure 9.** a-c) LOO diagnostics for points close (01), far (02) from the design location and large fluctuations in the results (03) (locations 01, 02 & 03 in Figure 4). The RMSE yields values that vary from 0.066 to 0.105 for locations 01 and 02 and 0.145 in location 03. d) Error estimation for the predictions of the last 10 runs at location 001.

The Root Mean Square Error (RMSE) is also relatively small ranging from 0.066 at location 01 to 0.145 at location 03 where the wave elevation is higher (Fig. 9a, b & c). The RMSE is computed at these three locations for illustrating the behavior of the emulator's predictions at certain points, the relative error might increase further inland at inundated locations. To gain a more comprehensive understanding of the RMSE trend and the efficacy of the design, we fit 4 emulators in location 01 using as training data the first 20, 30, 40 and 50 runs out of the 60 runs, predicting each time for the last 10 runs and calculating the RMSE (Fig. 9d). It is noticed that the error reduces to a significant extent when adding more runs to train the emulators and becomes very small for an emulator trained at 50 runs, following this trend, a smaller error would be also expected for the emulators trained at 60 runs (Fig. 9d). GP emulation is well suited for approximating nonlinear simulation behaviors, even when considering continuous outputs of low regularity and when restricted to small-sized experimental designs with space-filling properties. As shown by Owen et al. (2017), when considering two cases with computationally intensive simulators, more specifically, a land-surface simulator and a launch vehicle controller, GP emulation demonstrates good approximation properties even for small design sizes. By small design sizes, we refer to designs with the number of samples being about ten times the number of input parameters, a widely used rule of thumb for effective computer experiment design (Loeppky et al., 2009).

## 4.2  Predictions with two families of scenarios

Once the emulators are built, the maximum tsunami elevation can be predicted for any input deformation scenario. The prediction involves the utilisation of the built emulator with a given set of inputs to calculate the mean predictions and their uncertainty on the outputs. Uncertainties are fully propagated to display sometimes complex distributions of outputs such as skewed distributions (as it in the case below): variance would not be enough to describe such uncertainties. Emulation provides a complete description of uncertainties compared to a mean and variance in more basic approaches. These inputs can be represented by the distributions of the input parameters (Fig. 10). The distributions are flexible and can be used to represent different hypothetical cases. A beta distribution is employed for each parameter, from which 2,000 scenarios are randomly selected. The shape parameters of the distributions can be utilised to express the scientific knowledge on the source. To predict $H_{max}$ we initially explore the likelihood of maximum uplift to vary around 4 m, similar to the values inferred by Satake (2003) for the 1700 event, and within a range of 1-7 m (Fig. 10: $h_{max}$, H1). A maximum subsidence is considered with the minimum values to be more likely at 1/2 of the uplift (Fig. 10: $h_{min}/h_{max}$, H1). The total time of the event is considered to most likely vary around 300 s (Fig. 10: Total time, H1). For the most uncertain parameters we use a symmetric distribution. In more detail, the a & b shape parameters of the Beta distributions used to produce the 2,000 seabed displacement scenarios in Figures 11 and 12a & 12b have values of: Total Time: [2.5, 2], $Dh_{max}/Dh_{min}$: [2, 2], $Dd/Dh_{min}$: [2,2], $h_{min}/h_{max}$: [2,2.5], $h_{max}$: [3, 2] and $h_t/h_{max}$: [2, 2] (Fig. 10: H1).

The MOGP emulation framework allows us to produce the predictions of tsunami hazard in parallel for each emulator. The mean predictions are represented in the form of the histograms of the predictions at each location as shown in Figure 11 for locations 1, 2 & 3 of Figure 4. Locations 01 and 02 display very small elevation values above the ground level due to the tsunami. It appears that the splay-faulting sensitivity is larger at locations 1 and 3 than at location 2 since the histogram of

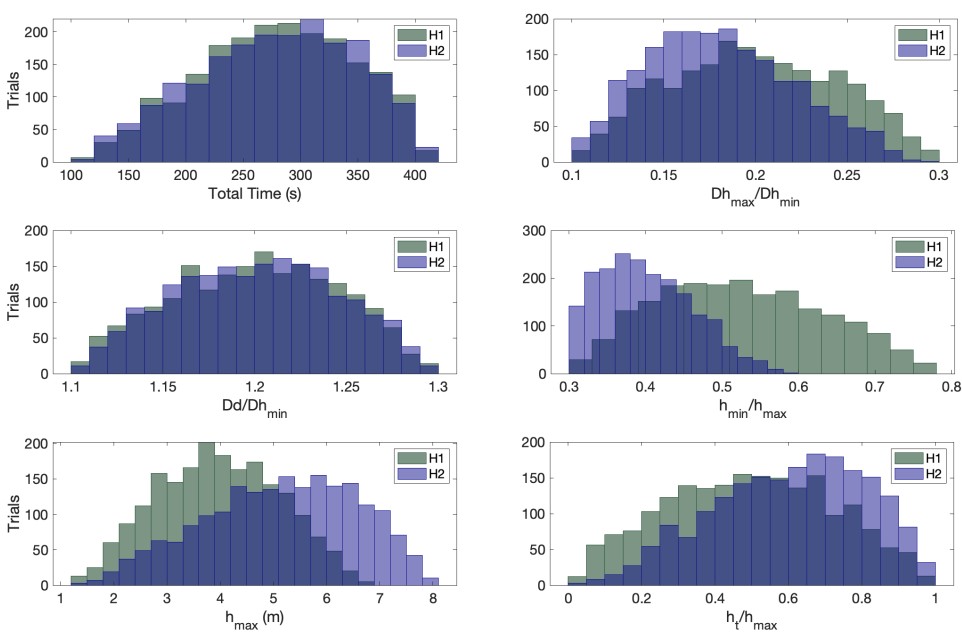

**Figure 10.** Input parameter distributions for two sets of hypothetical cases. Histograms of buried ruptures (H1) are depicted with dark green colour and of splay-faulting (H2) with dark blue colour.

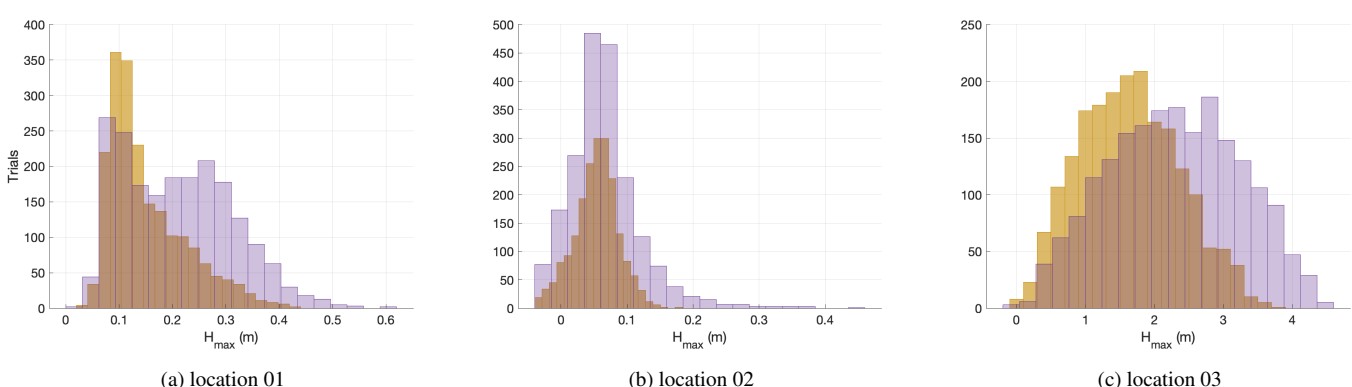

(a) location 01          (b) location 02          (c) location 03

**Figure 11.** Predictions for three cell-centres of the grid (see locations 01, 02 & 03 respectively in Fig. 4) for scenarios resulting from the distributions H1 (buried ruptures, prediction histograms in yellow) & H2 (splay-faulting, prediction histograms in purple).

outputs shifts more towards higher values. Location 03 gives values that most likely range around 1.5 m depending on the selected deformation scenarios. The minimum values shown in the histograms can become negative since a positive prediction
is not imposed by the emulator, but this is very rare and it does not manifest in the production of the hazard maps.

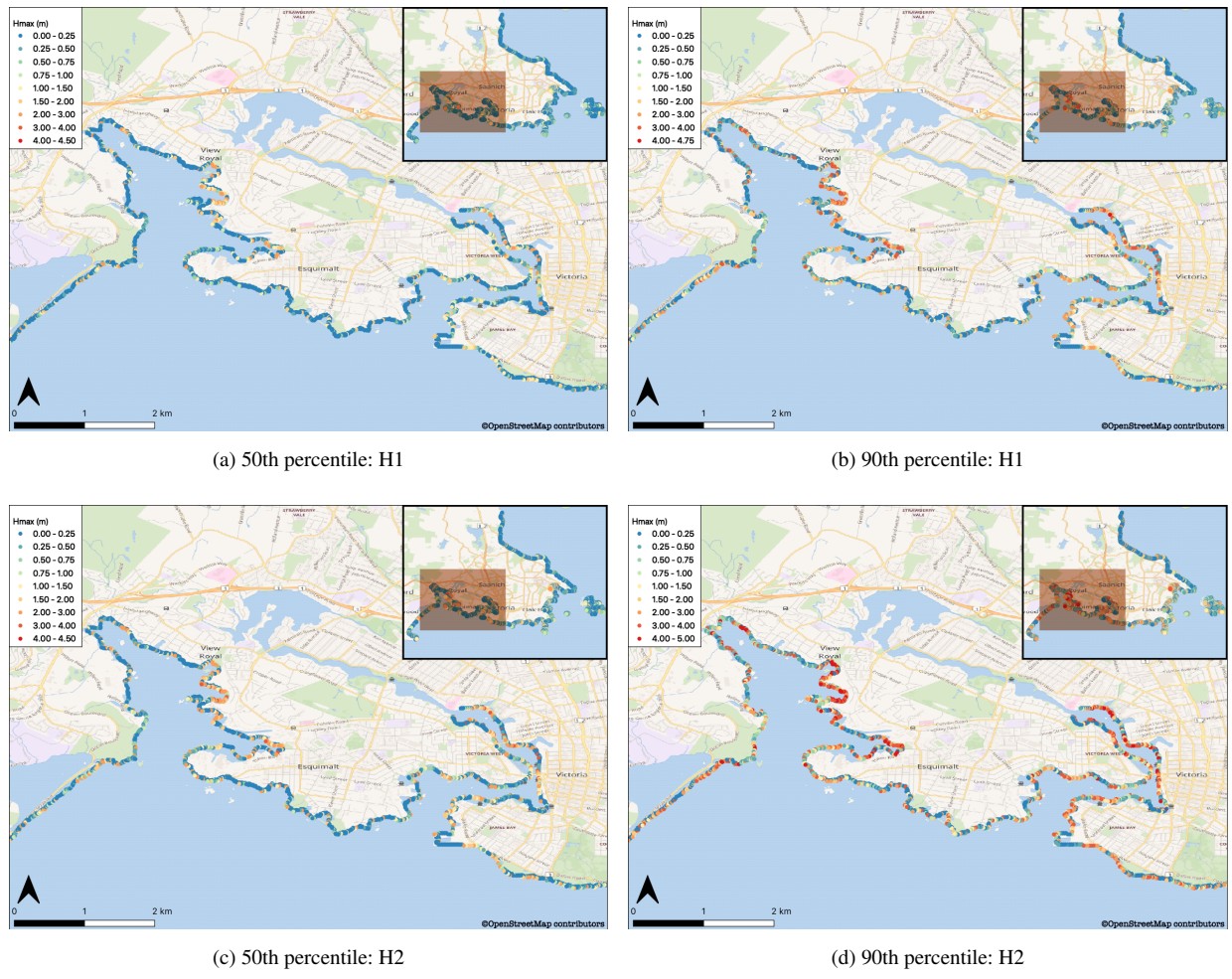

(a) 50th percentile: H1

(b) 90th percentile: H1

(c) 50th percentile: H2

(d) 90th percentile: H2

**Figure 12.** The percentiles (50th left, 90th right) of the mean predictions at the cell centres of the computational grid, for buried ruptures (H1) and for splay-faulting (H2). The circles show the locations of the emulators. The figures were produced with the QGIS software using as base-maps the Wikimedia layers with data provided by OpenStreetMap contributors, 2021. Distributed under a Creative Commons BY-SA License.

The hazard maps in the South-East part of Vancouver island are produced based on the 50th and 90th percentiles of the emulator's predictions for the 2,000 tsunami scenarios. 5,148 coastline locations that correspond to the cell-centres of the computational grid are studied (Fig. 12a & b). The 50th percentile of the predictions demonstrates that 67.19% of the predictions (3,459 locations) have values between 0 and 0.25 m, whereas 86.36% (4,446 locations) falls under 1 m (Fig. 12a). When considering the 90th percentile, however $H_{max}$ values increase. The results show that 48.77% of the predictions (2,511 locations) have $H_{max}$ between 0 and 0.25 m, 74.86 % (3,854) of the predictions falls under 1 m (Fig. 12b). It is observed that $H_{max}$ ranges between 1 - 3 m at 22.76% of the locations (1172 locations) (Fig. 12b).

It is noted that the fitting of each emulator takes approximately 1.5-3.0 s, whereas each emulator prediction takes ca. 0.001 s on the KNL nodes of the Cambridge High Performance Computer Service (Peta4-KNL of the CSD3 cluster). Hence, once the emulators are built, they can be used to explore alternative rupture scenarios in fast times. Such is the hypothetical case of increased uplift in the northern part of the subduction zone caused by splay-faulting. There is a large uncertainty surrounding the presence of splay-faulting in the Northern part of the zone (Gao et al., 2018). Witter et al. (2013) have estimated the probability of splay-faulting during a megathrust earthquake to be at ca. 60%. Although such enhanced vertical displacements are not likely to occur in the southern part of the zone, the tsunami impact from a short-north segment or a long rupture could be similar for British Columbia (Cherniawsky et al., 2007). To fully assess splay-faulting related tsunami hazard in Southeast Vancouver island, the complexity of the fault geometry needs to be more accurately incorporated at the initial stages of the process. The impact of an enhanced uplift is, thus, explored in a simplified form for the area here. Keeping the distributions of time and $Dd/Dh_{min}$ the same, 2,000 additional rupture scenarios are predicted to study the impact of a larger rupture in the region. Larger rupture scenarios may be characterised by an increased uplift and a more abrupt vertical deformation, the shapes of the parameter distributions for $Dh_{max}/Dh_{min}$, $h_{max}$ and $h_t/h_{max}$ can be, thus, defined by $a = [2, 3, 3]$ and $b = [3, 2, 2]$ respectively (Fig. 10: H2). These values raise the likelihood of the maximum uplift to vary between 5-7 m (Fig. 10: H2). The ratio of $h_{min}/h_{max}$ is estimated to be lower ($a = 1.5$, $b = 3$), as the maximum subsidence in worst-case rupture scenarios is expected to be at ca. -2 m (AECOM, 2013; Witter et al., 2013).

Looking at the 50th percentile of the predictions for H2, it is shown that the $H_{max}$ values from these scenarios are increased (Fig.12c). In Figure 12c, 57.98% (2.985 locations) of the predicted $H_{max}$ falls between 0.0-0.25 m, whereas 79.51% (4,093 locations) falls under 1 m. In both hypothetical cases the large majority of the predicted $H_{max}$ falls under 2 m (98.17% of the locations in Figure 12a and 94.09% in Figure 12c). However, when considering the 90th percentile of the predictions, the outputs become more severe (Fig. 12d). In this case, only slightly more than the one third of $H_{max}$ (35.18% of the locations) falls within the range of 0-0.25 m. $H_{max}$ falls below 1 m at 63.81% (3,285) of the locations, and ranges between 1-3m at 27.93% (1,438) of the locations. Maximum wave heights/flow depths between 3-4 m are recorded at 6.29 % of the locations (Fig. 12d) as opposed to 2.33% in the previous case (Fig. 12b). Following similar procedures, seismic data in combination with expert knowledge on the rupture characteristics can be translated to probabilistic tsunami hazard outputs.

Tsunami amplification is especially apparent at narrow bays and coves inside the Victoria and Esquimalt harbors and is likely the outcome of wave resonance (Fig. 12). Wave amplification in harbors and small bays has also been observed in other numerical studies in the area (Cherniawsky et al., 2007; AECOM, 2013; Fine et al., 2018). In their numerical studies of large

earthquake-induced tsunamis, Cherniawsky et al. (2007) found maximum elevations above 4 m at the Northwest shallow parts of Esquimalt harbour with the second wave peak being larger than the first one in some locations. Similar values (ca. 4.3 m) have been computed by AECOM (2013) in the area. These higher values are possibly the effect of wave resonance attributed to the regional geomorphology. Wave resonance has been observed in Port Alberni, located at the head of a narrow inlet in the west past of Vancouver island, during the 1964 Alaskan earthquake (Fine et al., 2008). The recorded wave heights in the port were 3-4 times larger than in the adjacent areas, often recorded in the third or later waves, and the tsunami oscillations continued for days after the event (Fine et al., 2008). It is likely that local topographic features can contribute to tsunami amplification also in other parts of the region.

### 4.3   Probabilistic tsunami hazard calculation

Further, we associate the scenarios with annual frequencies to be able to calculate probability of exceedance for predictions of the H2 distribution. We study the pattern of 1/1000 year exceedance rate flow depths over the region, drawing elements from the process followed in Park et al. (2017). The probability of a full-margin rupture generated tsunami is considered, the earthquake magnitudes associated with such an event important for the tsunami hazard assessment are in the range of $Mw = 8.7 - 9.3$. To link seabed deformations to earthquake moment magnitudes, we use a simplified approach by matching maximum seabed uplift, calculated using the Okada (1985) solution for idealized planar fault, with rupture dimensions similar to Cascadia subduction interface experiencing linearly decaying slip with depth. Following this approach, the magnitudes of the H2 scenarios range between $8.77 - 9.28$ (Fig. 13a). To associate frequency of events with earthquake magnitudes, a Tapered Gutenberg-Richter (TGR) distribution is utilized which has been proved to give adequate predictions for the Cascadia subduction zone (Rong et al., 2014). The TGR Complementary Cumulative Distribution function for a given earthquake magnitude $m$ is defined as:

$$F(m) = [10^{1.5(m_t - m)}]^\beta \exp[10^{1.5(m_t - m_c)} - 10^{1.5(m - m_c)}],$$

where $m_t$ is a threshold magnitude above which the catalogue is assumed to be complete (here $m_t = 6.0$), $m_c$ is the corner magnitude and $\beta$ is the index parameter of the distribution. Considering the 10,000-year paleoseismic record, as reconstructed by Goldfinger et al. (2012) from turbidite data, $m_c$ and $\beta$ take values of 9.02 and 0.59 respectively. The discrete number of $m_j$ magnitudes can also be computed by:

$$P[M = m_j] = G(m_j + 0.5\Delta m) - G(m_j - 0.5\Delta m),$$

where $G(m) = 1 - F(m)$ denotes the cumulative density function and $\Delta m$ is the discretization interval. Following the above, the number of earthquakes in 200K years per magnitude band can be estimated as shown in Fig. 13b. In the same figure, the number of events per magnitude band for the H2 predictions is shown. Because prediction parameters, especially $h_{max}$, were drawn from independent distributions, the frequencies of the H2 set events need to be re-scaled using the ratio between number

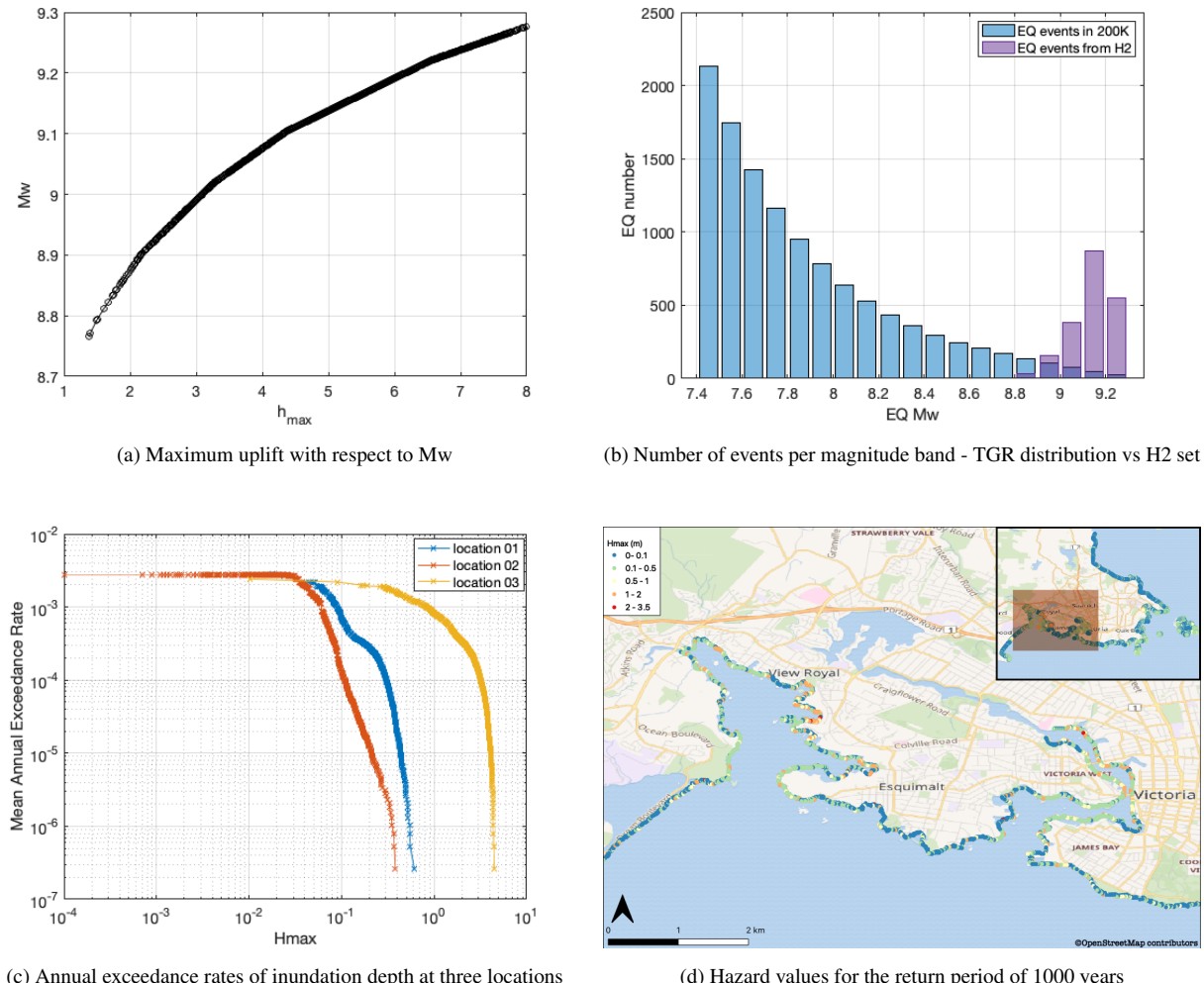

(a) Maximum uplift with respect to Mw

(b) Number of events per magnitude band - TGR distribution vs H2 set

(c) Annual exceedance rates of inundation depth at three locations

(d) Hazard values for the return period of 1000 years

**Figure 13.** Panel (a) shows the relationship between earthquake magnitude and maximum uplift when using a linear Okada solution for full rupture of the zone. Number of events (b) according to the Tappered Gutenberg-Richter distribution for a 200K year band (blue) and the H2 scenarios (purple). Sample hazard curves for three locations (c) and coastal hazard map for the 1000 years return period (d) the figure produced with the QGIS software using as base-maps the Wikimedia layers with data provided by OpenStreetMap contributors, 2021. Distributed under a Creative Commons BY-SA License.

of events per magnitude band for TGR distribution and H2 set to assign the appropriate relative frequency of each event within H2.

Having the event frequencies, occurrence exceedance probability for the hazard values can be calculated for each of the sites. We start by arranging the hazard values at a site in descending order: $h_1 > h_2 > \cdots > h_n$, with $n = 2000$. The exceedance probability for the largest hazard value $h_1$ (corresponding magnitude is Mw 9.2 - 9.3) becomes $P_{ex}(h_1) = 0$. For the second largest hazard value $P_{ex}(h_2) = f_1$, and for the other hazard values can be computed recurrently as:

$$P_{ex}(h_{i+1}) = 1 - (1 - P_{ex}(h_i))(1 - f_i) \qquad \text{for } i = 2, 3, \ldots, n,$$

The above relations are valid for a set of independent events when their annual occurrence rates are known. They are derived from basic probability theory and used in hazard analysis studies, such as for example in Monte Carlo event-based probabilistic seismic hazard assessment (e.g. Musson (2009)). Accordingly, the mean annual exceedance rate can be computed for the hazard values at each location (Fig. 13c). Considering then a 1/1000 exceedance rate, according to the hazard curves for locations 1-3 of Fig. 4, the most significant tsunami run-up is expected for location 3 (between 80-90 cm). Larger hazard values ( slightly above 3 m) are expected for location 3 when considering a 1/10,000 exceedance rate, whereas for the other two locations the values would fall below 25 cm (Fig. 13c). The hazard curves for each location can be used to construct the hazard map of Figure 13d, which represents the $H_{max}$ for the H2 events occurring in 1000 years. The hazard map shows that when considering an event within this interval, 4.19% of the locations have $H_{max}$ above 1 m. The majority of the locations experience $H_{max}$ below 1 m. Compared to the hazard values at Seaside, Oregon, as calculated by Park et al. (2017) for the 1/1000 probability of exceedance, the expected hazard at Victoria sites is significantly lower. One factor for these discrepancies is the location of the two sites, as Seaside is impacted by the tsunami waves from Cascadia subduction zone directly, whereas Victoria is protected by the Olympic Peninsula and the west side of Vancouver island, therefore, to reach sites in Victoria, the waves have to travel much further and are attenuated along their path.

We note that as a first demonstration on how the emulators' predictions can be linked with the probability of exceeding a tsunami intensity measure over time, this is a simplified case. To better capture the probabilistic tsunami hazard in the region the seabed deformation parameter distributions used for generation of the predictions need to be more precisely associated with Cascadia rupture characteristics. In future research, the Okada solution for a realistic slip distribution on precisely modelled Cascadia subduction interface will be employed to generate more physically driven seabed uplifts. To these uplifts, perturbations can be applied to gather a distribution of deformation parameters. Also, alternative more realistic magnitude-frequency relationships than the selected Tapered Gutenberg-Richter distribution can be considered, for example the distribution used by the Geological Survey of Canada for the new 2020 6th Generation of seismic hazard model for Canada or for the 2018 National Hazard for the United States.

## 5 Conclusions

In this work, a sequential design algorithm was employed for the conduction of the computational experiments for earthquake generated tsunami hazard in the Cascadia subduction zone. This approach aided an informative, innovative selection of the sets of numerical experiments in order to train the statistical emulators. It forms the first of its kind, to the authors' knowledge, which involves the application of a sequential design algorithm towards realistic tsunami hazard predictions through emulation. Focusing the high-resolution computations in the Southeast Vancouver island, $H_{max}$ was predicted at 5,148 coastal locations with the utilisation of the emulators. Once the emulators are built, expert knowledge can be facilitated to swiftly assess hazard in the region. The flexibility of the method allowed, here, to predict thousands of scenarios in a few moments of time under different parameter set-ups. The hazard outputs demonstrated in the study resulted from 2,000 potential rupture scenarios, the parameters of which were distributed following two hypothetical cases (2,000 predictions/case). The emulators' predictions were linked to their occurrence exceedance probability which allowed us to produce probabilistic hazard maps that assess the hazard intensity of such events in the area (Fig. 13). This forms one way of representing the mean predictions under a probabilistic framework. Alternatively one could present other probabilistic statements, for instance assessing the probability of exceeding some given threshold of maximum tsunami run-up. This methodology could prove useful for assessing the hazard at the first stages of mitigation planning in order to take preventive measures such as built structures or natural hazard solutions.

The predictions showed a high dependence of the maximum wave heights/flow depths on the maximum uplift during the rupture. In the areas of Victoria and Esquimalt, the majority of the predictions tend to be under 1 m and most likely under 0.25 m. However, wave amplification is observed inside the harbors and especially in narrow bays and coves, possibly as an effect of wave resonance. When considering the maximum uplift distributions with a higher likelihood ranging between 4-7 m, the 90th percentile of the predictions shows that $H_{max}$ ranged between 3-4 m at 6.29 % of the locations studied. In rare cases (at 1.9% of the locations) the values may exceed the threshold of 4 m, falling within a range of 4-4.9 m. Similarly for the probabilistic tsunami hazard for 1/1000 exceedance rate, 4.19% of the locations experience $H_{max}$ above 1 m. These percentages are expected to increase when looking at larger return periods and the hazard values have to be further assessed to produce a probabilistic risk assessment for the area.

This study expands on the methodology and the development of the workflow to build the emulators under a sequential design approach. As so, there are some aspects that need to be considered in future work to further refine the probabilistic outputs. These span from the tsunami generation to the inundation. In this case, an idealised geometry was used for the source, and the current results agree with the numerical studies of more incorporating fault geometries. However, to fully explore the complexity of the rupture, future work would benefit from the integration of compound rupture characteristics, especially when it comes to splay-faulting consideration. Furthermore, gaps and mismatches in the digital elevation data should be accounted for and incorporated in the modelling for a more finely resolved representation. Uncertainties and/or errors in the bathymetry and elevation data may play a critical role in the wave elevation outputs when assessing tsunami impact at a high-resolution and can be included in the emulation (Liu and Guillas, 2017). Model bias is also not addressed in this study, but could be explored in future investigations, for example by correcting the bias by adding a discrepancy estimated by comparing against

past observations. Finally, to produce a complete hazard assessment in the region, probabilistic tsunami inundation should be carried out. This is enabled by highly nonlinear features in the emulators' predictions and even benefit from recent advances in emulation (Ming et al., 2021) whenever nonlinearities consist of dramatic step changes, e.g. in the case where over-topping of defenses would generate vastly different flooding patterns.

*Code availability.* MOGP-MICE: https://github.com/alan-turing-institute/mogp_emulator, VOLNA-OP2: https://github.com/reguly/volna.git, https://github.com/DanGiles/volna.git

**Appendix A**

Table A1: Deformation scenarios selected by MICE to be used as source in the tsunami simulations.

| Scenario | Time (s) | $Dh_{max}/Dh_{min}$ | $Dd/Dh_{min}$ | $h_{min}/h_{max}$ | $h_{max}$ (m) | $h_t/h_{max}$ |
|---|---|---|---|---|---|---|
| 01 | 415 | 0.18876 | 1.14439 | 0.65090 | 3.57400 | 0.89827 |
| 02 | 118 | 0.11066 | 1.18921 | 0.52726 | 2.25844 | 0.51710 |
| 03 | 322 | 0.26905 | 1.29754 | 0.40622 | 5.15068 | 0.24049 |
| 04 | 269 | 0.15534 | 1.24382 | 0.79471 | 6.92122 | 0.05661 |
| 05 | 168 | 0.25932 | 1.10375 | 0.36200 | 6.27256 | 0.73954 |
| 06 | 324 | 0.19257 | 1.13175 | 0.58639 | 1.20224 | 0.78907 |
| 07 | 333 | 0.29182 | 1.29837 | 0.46602 | 1.14988 | 0.36028 |
| 08 | 321 | 0.20928 | 1.11077 | 0.78350 | 1.07064 | 0.68148 |
| 09 | 415 | 0.28950 | 1.14091 | 0.31510 | 1.24523 | 0.87016 |
| 10 | 396 | 0.28166 | 1.26802 | 0.42280 | 1.32406 | 0.98242 |
| 11 | 203 | 0.28396 | 1.10121 | 0.40705 | 7.95143 | 0.95837 |
| 12 | 142 | 0.13749 | 1.11697 | 0.54020 | 7.95486 | 0.98510 |
| 13 | 346 | 0.17685 | 1.12009 | 0.47852 | 7.80322 | 0.84656 |
| 14 | 173 | 0.13373 | 1.23984 | 0.34994 | 7.98143 | 0.69699 |
| 15 | 336 | 0.23286 | 1.27498 | 0.55781 | 7.91868 | 0.67009 |
| 16 | 237 | 0.13273 | 1.27853 | 0.54955 | 5.63092 | 0.98188 |
| 17 | 343 | 0.12069 | 1.13049 | 0.53715 | 2.21708 | 0.01722 |
| 18 | 412 | 0.10322 | 1.29979 | 0.36434 | 2.54273 | 0.07739 |
| 19 | 120 | 0.29840 | 1.18448 | 0.73063 | 3.40402 | 0.22795 |
| 20 | 199 | 0.12152 | 1.10658 | 0.76242 | 5.12636 | 0.97087 |
| 21 | 250 | 0.20656 | 1.12431 | 0.38110 | 3.47216 | 0.41432 |

| 22 | 106 | 0.15650 | 1.19044 | 0.37480 | 3.91755 | 0.76439 |
|----|-----|---------|---------|---------|---------|---------|
| 23 | 227 | 0.19896 | 1.18189 | 0.39482 | 3.34181 | 0.06342 |
| 24 | 281 | 0.18439 | 1.18176 | 0.31049 | 4.09125 | 0.36047 |
| 25 | 132 | 0.22202 | 1.17230 | 0.32744 | 4.04066 | 0.50855 |
| 26 | 403 | 0.10581 | 1.15201 | 0.32755 | 5.95859 | 0.61608 |
| 27 | 121 | 0.25499 | 1.22249 | 0.74695 | 6.03230 | 0.95641 |
| 28 | 109 | 0.22550 | 1.29511 | 0.76429 | 5.62019 | 0.93289 |
| 29 | 398 | 0.11921 | 1.25359 | 0.45199 | 6.48259 | 0.53838 |
| 30 | 137 | 0.26685 | 1.20942 | 0.77697 | 6.98940 | 0.85119 |
| 31 | 148 | 0.27108 | 1.12253 | 0.56776 | 6.50266 | 0.01205 |
| 32 | 163 | 0.27646 | 1.14596 | 0.78362 | 2.18301 | 0.97280 |
| 33 | 159 | 0.27577 | 1.11221 | 0.32033 | 2.62842 | 0.86107 |
| 34 | 224 | 0.28072 | 1.23412 | 0.77180 | 2.09684 | 0.98993 |
| 35 | 213 | 0.10919 | 1.19239 | 0.64663 | 4.66165 | 0.09996 |
| 36 | 361 | 0.19797 | 1.10408 | 0.62892 | 5.58353 | 0.34304 |
| 37 | 128 | 0.18069 | 1.28457 | 0.30279 | 1.01087 | 0.08392 |
| 38 | 394 | 0.15899 | 1.11494 | 0.44509 | 7.50061 | 0.01201 |
| 39 | 413 | 0.27468 | 1.15351 | 0.57142 | 5.35065 | 0.23606 |
| 40 | 119 | 0.14168 | 1.19017 | 0.36867 | 1.09176 | 0.34923 |
| 41 | 410 | 0.20819 | 1.11398 | 0.70190 | 7.89979 | 0.56523 |
| 42 | 132 | 0.10632 | 1.20853 | 0.78520 | 3.21606 | 0.59659 |
| 43 | 130 | 0.29382 | 1.10213 | 0.50885 | 4.80543 | 0.69270 |
| 44 | 139 | 0.28712 | 1.10695 | 0.62522 | 1.68192 | 0.43708 |
| 45 | 361 | 0.25899 | 1.29084 | 0.45390 | 3.75642 | 0.93926 |
| 46 | 202 | 0.16599 | 1.24416 | 0.76199 | 2.22083 | 0.02408 |
| 47 | 141 | 0.10912 | 1.27527 | 0.36927 | 6.49197 | 0.02806 |
| 48 | 234 | 0.26551 | 1.29713 | 0.73814 | 1.38065 | 0.03826 |
| 49 | 111 | 0.11068 | 1.29673 | 0.55071 | 6.42067 | 0.26797 |
| 50 | 120 | 0.18138 | 1.29293 | 0.63737 | 1.18291 | 0.61683 |
| 51 | 376 | 0.14548 | 1.20031 | 0.79548 | 6.01218 | 0.97036 |
| 52 | 211 | 0.28749 | 1.21302 | 0.45755 | 6.80059 | 0.32854 |
| 53 | 157 | 0.21934 | 1.21763 | 0.48670 | 7.53270 | 0.41969 |
| 54 | 115 | 0.15668 | 1.14711 | 0.71566 | 7.81272 | 0.03208 |
| 55 | 375 | 0.20861 | 1.27150 | 0.66456 | 4.61712 | 0.26980 |
| 56 | 369 | 0.14922 | 1.27388 | 0.30216 | 2.26079 | 0.90427 |

| 57 | 340 | 0.12888 | 1.20553 | 0.34914 | 2.87207 | 0.80172 |
| 58 | 363 | 0.27871 | 1.26487 | 0.55128 | 2.49722 | 0.52515 |
| 59 | 376 | 0.25478 | 1.12818 | 0.76070 | 2.63381 | 0.65489 |
| 60 | 405 | 0.29587 | 1.24347 | 0.68209 | 2.90920 | 0.56893 |

*Author contributions.*  DS carried out the analysis and writing. JB, PP and SG contributed to the analysis and writing. SG also supervised the
analysis.

*Competing interests.*  The authors declare that they have no conflict of interest.

*Acknowledgements.*  We gratefully acknowledge the support, advice and fruitful discussions with Dr Simon Day. His help was important
in the first stages of the study for the conceptualisation of the seabed deformation and the parameterisation. We are also thankful to Dr
Devaraj Gopinathan, for his support with the meshing techniques for the numerical simulations. We also thank two anonymous reviewers
for their thorough comments which greatly helped us to improve and clarify the manuscript. Finally, we would like to thank the Research
Software Engineering team at the Alan Turing Institute, and especially Dr Eric Daub and Dr Oliver Strickson for their help with MOGP and
the automation of the workflows. SG was supported by the Alan Turing Institute project "Uncertainty Quantification of complex computer
models. Applications to tsunami and climate" under the EPSRC grant EP/N510129/1.

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
