# Peer review of "Probabilistic, high-resolution tsunami predictions in North Cascadia by exploiting sequential design for efficient emulation"

_Natural Hazards and Earth System Sciences, 2021_

## Author Comment (AC1)

Referee Report for

Title: Probabilistic, high-resolution tsunami predictions in North Cascadia

by exploiting sequential design for efficient emulation

Author(s): Dimitra M. Salmanidou et al.

MS No.: nhess-2021-63

MS type: Research article

Referee Report #1

GENERAL COMMENTS

Dear authors,

This study implements and describes the results of the application of an emulator for the modelling of tsunamis generated by co-seismic seafloor displacement, which produces a very satisfying trade-off between accuracy and computational cost. This approach can be advantageous for tsunami hazard analysis typically requiring computationally intensive exploration: 1) of the expected natural variability (the aleatory uncertainty), and 2) of different yet still credible modelling choices consistent with - limited - observations (the epistemic uncertainty).

I consider the technical topic dealt with here of great relevance. Saving computational resources allows in principle their more efficient usage to quantify and even reduce uncertainties through a smarter exploration of the parameter space characterising the natural phenomena under scrutiny.

However, I see at least three issues that need to be addressed:

1. the limited originality or the need for a better framing with respect to the state of the art;
2. the weakness of some elements of the modelling approach;
3. the overstatement of the focus/message.

For this reason, I suggest that the paper undergoes major revisions before being published.

In what follows, I explain the three issues listed above.

We would like to kindly thank the reviewer for this in-depth review that helped us to improve the clarity and the message of our manuscript. Guided by their comments, we decided to expand on the probabilistic aspect of our work, which we think also increases the novelty of our manuscript. We provide a point-by-point response below.

1) Originality.

It is not completely clear to me how much this paper presents a definite advancement over previous work, mostly by some of the authors of this study.

The authors state at lines 9-10 in the Abstract: "This approach allows for a first emulation-accelerated computation of probabilistic tsunami hazard in the region of the city of Victoria, British Columbia."

Actually, the one presented is far from being a probabilistic tsunami hazard assessment, as detailed in point 3) below.

Nevertheless, this might still be a methodological paper regarding some aspects of the tsunami hazard assessment workflow.

However, the authors - or some of them I'm not sure - already beautifully clarified the potential impact of their emulator in combination with the VOLNA simulator.

In fact, they say (lines 44-45): "Statistical emulators (also known as a statistical surrogate models) can be called to address these issues (Behrens and Dias, 2015)."

And also: (lines 46-48): "Such approaches have been implemented for uncertainty quantification of tsunami hazard at various settings (Sraj et al., 2014; Salmanidou et al., 2017, 2019; Guillas et al., 2018; Denamiel et al., 2019; Snelling et al., 2020; Gopinathan et al., 2021).

Then, I'm struggling to understand the real advancement here. I see this work mostly as a kind of reshuffling of previous methods, for example, the one used in Gopinathan et al. (2021), which relies on Beck and Guillas (2016), or the one applied by Giles et al. (2021, https://doi.org/10.3389/feart.2020.597865), not cited here, which uses the Gaussian Process emulator code but not, if I get it right, the adaptive sequential design.

But maybe there are technical details that really represent a leap ahead and just deserve to be better highlighted.

So, let me assume for a moment that this is not the case and the novelty brought by this study can be shown beyond any reasonable doubt: even so,

1. the description of the relationship of the present work with the existing literature should be improved (see specific comments);
2. the advantage over the previously applied techniques should be clearly demonstrated by means of the use-case, it cannot be only declared based on the author's experience.

Conversely, if a novel significant methodological development was not achieved, and the study is only the application of an existing yet pretty sound technique to a slightly different use-case, the paper may lack enough scientific significance.

Please, address this first issue with great care.

Thank you very much for your comment. The novelty here is the use of the sequential design MICE by Beck and Guillas for the construction of the GP emulator of the tsunami model. This is done for the very first time towards a realistic case using High Performance Computing. Gopinathan et al (2021) and Giles et al. (2021) (now also cited in the revised manuscript) used a one shot random sampling for the training which lacks the information gain achieved by sequential design. Concretely, sequential design can reduce by 50% the computational cost, as demonstrated in (Beck, Guillas, 2016) for a set of toy problems, so applying the approach towards a realistic case is showcasing real benefits in the case of high resolution with a complex parametrization of the source and is new. Note, that Guillas

et al. (2018) was neither using high resolution nor sequential design and the source was much simpler. The relationship of this work with existing literature is given below in the answers to specific comments.

2) Modelling.

Even if, overall, the approach is certainly up to the best international standards, and probably even pushing the limit a bit beyond, it remains very weak if not insufficient in certain parts and aspects of the proposed workflow.

The approach to the modelling of the seafloor co-seismic displacement is too simplistic and completely subjective. There's a bunch of methods out there, some of which have been used for decades now, ranging from very simple modelling techniques to much more sophisticated ones.

For example, Gopinathan et al. (2021), with one of the coauthors of this study, apply a much more realistic approach.

In general, the different degree of complexity used by these methods may be chosen to match the "amount" of underlying physics, geology, and available constraints, for example deriving from the understanding of the regional seismotectonics and/or seismic and tsunami history.

As a term of comparison, among others, a couple of quite popular reviews focussing on seismic source modelling for tsunami application are

Geist, E. L., and Oglesby, D. D. (2014). Earthquake mechanism and seafloor deformation for tsunami generation, in Encyclopedia of earthquake engineering. Editors M. Beer, I. A. Kougioumtzoglou, E. Patelli, and I. S.-K. Au (Berlin, Heidelberg: Springer), 1–17. doi:10.1007/978-3-642-36197-5_296-1

Geist, E. L., Oglesby, D. D., and Ryan, K. J. (2019). "Tsunamis: stochastic models of occurrence and generation mechanisms," in Encyclopedia of complexity and systems science," Editor R. A. Meyers (Berlin, Heidelberg: Springer), 1–30. doi:10.1007/978-3-642-27737-5_595-2

There are two options that I foresee to improve the current situation. Either you make your choice and revise the modelling approach, or at least state clearly up front that you focus on different aspects than the earthquake modelling in this paper, and as a consequence prefer using a very simplistic approach for illustrative purposes only. It should also be stated very clearly that this approach should be replaced by a more realistic one for any real application.

Yet, how much the (over-)simplifications introduced here affect the characteristics of the resulting tsunamis remains to be addressed. Wave features in fact interact with bathymetry and topography features during tsunami evolution and they can strongly influence the inundation characteristics and extent. Different features of the waves may also result in a different performance of the emulator. These aspects should be discussed upfront in the paper as caveats for the readers to help them understand the real limitations of this approach.

We agree that the modelling of the seafloor co-seismic displacement is simplified in this study and it has to be improved in our future publications. This is mentioned in the

manuscript, but to improve clarity and avoid any confusion for the reader we add in the introduction:

"The focus of our work is on the methodological aspect of building the emulators and using them for multi-output tsunami hazard predictions. For a comprehensive tsunami hazard assessment, realistic modelling of Cascadia subduction interface magnitude-frequency relationships and seabed deformation parameterisation needs to be incorporated."

However, we argue that this parametrization of the seabed deformation can produce shapes that tend to be realistic e.g. when compared with one of the author's recent published work (Gopinathan et al. 2021 using lone Gaussian bumps). Our parametrization tends also to be more realistic than with stochastic models which may face different issues, e.g. creating negative slips to be corrected with ad hoc solutions (see also Davies et al., 2015). Such stochastic slips models also cannot accommodate an efficient parametric representation for emulation.

We agree it is difficult to estimate how sensitive the output of PTH can be to the set of seabed deformation shapes used in the modelling - this is not the topic of the present study and remains a topic of future research, however we do compare the results of a scenario with the results of similar studies in section 3.1 to ensure that they fall within reasonable ranges.

Davies, G., Horspool, N., & Miller, V. (2015) "Tsunami inundation from heterogeneous earthquake slip distributions: Evaluation of synthetic source models. *Journal of Geophysical Research: Solid Earth*, *120*(9), 6431-6451.

3) Focus/Message: this is not a probabilistic hazard assessment, rather it is an illustration of a method to deal with a component potentially useful for a hazard assessment.

The title and some descriptions in this paper may be confusing since one may think it deals with probabilistic tsunami hazard analysis, which is not true in my opinion. This paper, as stated at the very beginning of this report, deals with a technique for reducing computational cost. This technique is potentially applicable for hazard analysis. So, this ambiguity and some related overstatements should be fixed.

Let me try and clarify this concept. Can the emulation (or simulation) of an arbitrarily chosen set of scenarios be called "probabilistic tsunami hazard" (as for example in the abstract, at line 9)? What is a "probabilistic tsunami hazard"?

In its most common acceptance, a probabilistic tsunami hazard analysis provides the probability of exceedance in a given time interval of different thresholds of the chosen hazard intensity at a specific location. In general, this is made in the following way. Even if limited only to tsunamis generated by earthquakes, an attempt is typically made to take into account the full range of earthquake (parameter) variability in the hazard assessment; then, a model of their temporal occurrence, combined with the effect of each modelled tsunami scenario, allows the tsunami probability to be estimated. Otherwise, modelling a subjectively chosen range of scenarios, maybe similar to some historical events, is a what-if experiment, addressing the consequences of the hypothesised set.

On the contrary, in this study, a set of scenarios is quite arbitrarily chosen; then a beta distribution is assigned from which the scenarios to be modelled are sampled (Fig. 11). It is correctly stated that "The shape parameters of the distributions can be utilised to express the scientific knowledge on the source" (lines 254-255), but no effort is made to link the parameters of this distribution to reality, except for a generic correspondence of the

maximum values of the uplift-subsidence to those estimated for historical events or previous studies. This is a scenario analysis, based on a quite generic parameter setting, not a full probabilistic hazard assessment. This is perfectly fine for illustrative purposes of the potential usage of the emulation technique, provided that things are correctly framed.

So, please, improve the description throughout the entire manuscript to resolve this ambiguity, by eliminating any attempt to call it probabilistic hazard analysis.

Many additional comments related to the 3 points above are described in the specific comments in the following of this report.

We agree and we refrained from using the explicit term PTHA for our work. The emulators can give the probability of a tsunami intensity measure to exceed a certain threshold given predefined metrics at any horizon. We now provide a correspondence between predictions and time horizons. We make a first attempt here, which is simplistic but we believe adds to the novelty of the manuscript as it is, to our knowledge, the first to incorporate emulators for probabilistic tsunami hazard (as the term is currently accepted). The following section will be added and will be discussed in more detail in the revised manuscript:

"Further, we associate the scenarios and predictions of the H2 distribution with probability of exceedance. We study the pattern of 1/1500 year exceedance rate wave heights over the region drawing elements from the process followed in Park et al. (2017). The probability of a full-rupture generated tsunami is considered, the earthquake magnitudes associated with such an event are in the ranges of Mw=8.7-9.3 (Goldfinger et al., 2012). We link the moment magnitude for each event with the maximum seabed uplift caused by the linearly decaying slip on the rupture surface similar to the Cascadia subduction interface, a simple linear solution, first introduced by Okada (1985) is used. Following this approach, the magnitudes of the H2 scenarios range between 8.77-9.28. To associate frequency of events with earthquake magnitudes, a Tapered Gutenberg–Richter (TGR) distribution is utilized which has been proved to give adequate predictions for the Cascadia subduction zone (Rong et al., 2014). The TGR Complementary Cumulative Distribution function for a given earthquake magnitude $m$ can be computed as:

$$F(m) = \left[10^{1.5(m_t - m)}\right]^{\beta} exp\left[10^{1.5(m_t - m_c)} - 10^{1.5(m - m_c)}\right],$$

where $m_t$ is a threshold magnitude above which the catalogue is assumed to be complete (here $m_t = 6.0$), $m_c$ is the corner magnitude and $\beta$ is the index parameter of the distribution. Considering the 10,000-year paleoseismic record, as reconstructed by Goldfinger et al. (2012) from turbidite data, $m_c$ and $\beta$ take values of 9.02 and 0.59 respectively. The discrete number of $m_j$ magnitudes can also be computed by:

$$P[M = m_j] = G(m_j + 0.5\Delta m) - G(m_j - 0.5\Delta m),$$

where $G(m) = 1 - F(m)$ denotes the cumulative density function and $\Delta m$ is the discretization interval. Following the above, the mean annual rate of exceedance and the number of earthquakes in 200K years can be estimated as shown in Figure 13a. These relationships can then be used to associate the probability of exceedance for each event of the event set H2 by computing the relative frequency of each event within the data (Fig.13b). The occurrence exceedance probability for the hazard values $h_1 > h_2 > \cdots > h_n$, with $n = 2000$ becomes $P_{ex}(h_1) = 0$ for $h_1$ (corresponding magnitude is equal to 9.3), $P_{ex}(h_2) = f_1$, and for the other hazard values can be computed as:

$$P_{ex}(h_{i+1}) = 1 - (1 - P_{ex}(h_i))(1 - f_i) \quad \text{for} \quad i = 2,3 \ldots, n,$$

where $f_i$ is the event frequency assigned according to the magnitude bin to which the event was linked by its maximum uplift. The above relation comes from basic probability theory and is often used e.g. in Monte Carlo probabilistic seismic hazard assessment (Musson, 2000). Accordingly, the mean annual exceedance rate can be computed for the hazard values at each location (Fig. 13c). Considering then a 1/1500 exceedance rate, the hazard curves of each location can be used to construct the hazard maps of Figure 13d which represent the maximum wave heights from the H2 events.

[Figure]

Figure 13

We note that as a first demonstration of how the emulators' predictions can be linked with the probability of exceeding a tsunami intensity measure over time, this is a simplified case. To better capture the probabilistic tsunami hazard in the region the seabed deformation parameter distributions used for generation of the predictions need to be precisely associated with Cascadia rupture characteristics, other more realistic magnitude-frequency relationships than the selected Tapered Gutenberg–Richter distribution can be considered in future research."

SPECIFIC COMMENTS

Line 2:

"Traditional"? what do you mean by traditional? That the PTHAs for Cascadia have either low-resolution or use few scenarios? or that this is true all over the world?

I'm aware of several studies combining many scenarios at high resolution. Please, make a better survey of the relevant scientific literature.

We agree that, at some high cost, a large number of simulations can be carried out. For instance, Goda et al. 2020 generated 1000 tsunami simulations, but only below 90 m over a local area. We cannot afford to run hundreds of simulations at 30 m resolutions over Cascadia on a national-level GPU cluster where we were awarded 100,000 GPU-hours by the UK *Engineering and Physical Sciences Research Council* (*EPSRC*) to support several projects.

Hence, we corrected for the following sentence: "Previous probabilistic tsunami hazard assessment studies produced hazard curves based on simulated predictions of tsunami waves, either at low resolution, or at high resolution for a local area or under limited ranges of scenarios, or at a high computational cost to generate hundreds of scenarios at high resolution."

Goda, K., Yasuda, T., Mori, N., Muhammad, A., De Risi, R., and De Luca, F.: Uncertainty quantification of tsunami inundation in Kuroshio, Kochi Prefecture, Japan, using the Nankai–Tonankai megathrust rupture scenarios, Nat. Hazards Earth Syst. Sci., 20, 3039–3056, https://doi.org/10.5194/nhess-20-3039-2020, 2020.

Line 2:

Hazard curves, perhaps, not "hazard maps"; the PTHA produces hazard curves, from which the hazard maps can be extracted.

Corrected.

Line 3:

By "cost", do you mean: "computational cost"?

Actually, as stated above, there are several recent studies using many scenarios at high resolution. Mostly limited to one specific site indeed.

So, there is at least a third variable to consider, further than the range of scenarios, and resolution: the extent of the target coastal stretch.

Having said that, it is always a good idea to save resources, to be able then to expand in the right direction if needed.

However, it should be also mentioned that there are several recent approaches to deal with an optimal trade-off between cost and quality of the analysis.

So, this whole sentence should be reconsidered. This comment is connected to another one below on the need for citing relevant literature in the Introduction.

This sentence has been modified in response to the previous comments.

Line 9:

As already explained above, this cannot be considered a probabilistic tsunami hazard analysis.

Please see our response above.

Lines 30-35:

I suggest mentioning the global hazard assessment of Davies et al. and the Seaside hazard assessment of Gonzalez et al., both addressing PTHA at different resolution for the same target zone.

Thank you for this comment. The citations were added.

Lines 39-41:

It would be appropriate to mention, discuss, and cite the different existing techniques aimed at computational resource optimisation such as the class of methods based on the similarity between scenarios (e.g. offshore wave matching, cluster analysis), on importance sampling, and most recently on deep learning.

We add: "Other surrogate model techniques have been applied for tsunami or tsunami-like applications, such as polynomial regression (see, e.g., Kotani et al., 2020) and artificial neural networks (see, e.g., Yao, et al., 2021). For example, Yao et al. (2021) predicted tsunami-like wave run-up over fringing reefs using a neural network approach for approximating the relationship between inputs, including incident wave height and four reef features, and a wave run-up output on the back-reef beach. The authors emphasized that a disadvantage of artificial neural networks is that they are not suitable for small datasets. Owen et al. (2017) demonstrated, by examples involving computer-intensive simulation models, that GP emulation can approximate outputs of nonlinear behaviour with higher accuracy than polynomial regression when considering small- to moderate-sized, space-filling designs."

We also cite the computational resource optimisation of methods based on the similarity between scenarios (offshore wave matching, cluster analysis), e.g in Volpe et al. 2019.

Kotani, et al. (2020) "Probabilistic tsunami hazard assessment with simulation-based response surfaces" Coastal Engineering, 160: 103719.

Volpe, M., Lorito, S., Selva, J., Tonini, R., Romano, F., & Brizuela, B. (2019). From regional to local SPTHA: efficient computation of probabilistic tsunami inundation maps addressing near-field sources. *Natural Hazards and Earth System Sciences*, *19*(3), 455-469.

Yao, et al. (2021) "Predicting tsunami-like solitary wave run-up over fringing reefs using the multi-layer perceptron neural network", Natural Hazards, 107: 601-616.

Line 42:

I believe that it is not appropriate to state that high-resolution studies are necessarily needed for coastal planning. There are several meaningful - in my opinion - strategies based on lower resolution assessments. They are either approximations or stochastic treatments, or both, of the inundation probability, which are being used for evacuation or long-term coastal planning in different countries. Please, rephrase.

We agree, what we attempt to say is that in low resolution studies errors introduced in the geospatial data can introduce mistakes in the wave heights and inundation patterns. We modified the sentence to:

"High accuracy, high resolution computations are especially useful in tsunami modelling studies to assess inundation, damage to infrastructure and asset losses, but also for evacuation modelling."

Line 44:

About how "large" and "unaffordable", is it possible to provide some numbers? Otherwise, this would remain quite vague. What about using some references here?

We rephrase: "... the number of expensive numerical simulations needed to resolve the statistics about the output tends to be large (in the order of thousands for a well approximated distribution, Salmanidou et al., 2017, Gopinathan et al., 2021) and hard to materialise as it depends on the available resources, code architecture and other factors."

Line 45:
Probably, also Sarri et al. (2012) can be cited in this context.

Citation added.

Lines 47-48:
Also, Giles et al. (2021) can be probably cited here. However, as said above, the progress with respect to this list of papers should be clarified.

Citation added in the introduction.

Line 49:

For the sake of self-consistency, a short intro on what emulators are and how they work, and the role of training data would be useful.

We add: "Statistical emulators are stochastic approximations of the deterministic response. They are used to predict the expected outputs of the response at untried inputs that fall within the prescribed input parameter intervals. Training data, which are the observations of the response at various settings, are used to build the emulators and are thus of paramount importance. "

Line 52:

This comment on interpolation/extrapolation is crucial in my opinion, can you quantitatively prove your statement or at least qualitatively describe the reason in some detail?

We add:"Extrapolation means predicting outcomes for parameter values beyond the parameter domain on which emulators are designed to interpolate. Since the points representing seabed deformation scenarios are in a bounded parameter domain, emulators can mitigate undesired extrapolation if built on a training design set with good coverage of the domain, particularly if the envelope of the design set is close to the domain boundary. For small design sets, which we consider in this work, sequential design strategies are advantageous as they update the design set to improved coverage, among other desired design features, by conditioning on the current design point locations."

Lines 63-64:

Please add a reference for MOGP

The link to the open access GitHub repository of the code has been added as a footnote.

Lines 90-125:

This Section is inadequate for a real PTHA, as already explained.

The natural variability of earthquakes as currently understood is not represented. Nor are their temporal occurrence features.

The ones used here are kind of toy scenarios for tsunami initial conditions, and this makes the present study a "what-if" hazard assessment, which can have its own importance depending on the context, but it is not a full PTHA at all, for which among the most important aleatory variables are those describing the earthquake kinematics and/or mechanics, and the characteristics of the seismic cycle, along with the analysis of the epistemic uncertainty related to our understanding and to the (lack of) data available regarding the seismicity.

Your approach in my understanding is as follows: define an arbitrary shape related to a vague notion of how a co-seismic displacement profile would look like, parameterise it and vary these parameters within the ranges inferred by the characteristics of a few past earthquakes.

Please, use modern geology, physics and seismology, plus all available observations to constrain and validate your model, making it hopefully more realistic, while quantitively dealing with the large uncertainty involved.

Data and deep knowledge of the phenomena may help to limit subjectivity.

Alternatively, you may say that despite you are not dealing with PTHA and you are not using a state-of-the-art probabilistic earthquake model, you are proposing a method to deal with a specific segment of the analysis, aimed at reduction of the computational cost of the scenario simulation phase.

We understand the concerns raised by the reviewer and the focus of the manuscript has not been to do a PTHA for the region. We hope we addressed some of these concerns with our previous replies.

Line 94:

These by Stake and Wang are not two different methods for representing the seabed displacement.

The first one has to do with the reconstruction of a single event. The second one deals with the interseismic period, in turn indirectly linked to the slip, which causes the displacement. Unclear why both of them are indicated here as paradigmatic examples of the tsunamigenic displacement on this subduction zone.

Corrected.

Line 95:

The function used is not smooth, as I understand it.

Moreover, it is not completely clear how the full-length "rupture" is built and what's the temporal history of this "rupture".

We agree with your observation of the function not being smooth. Indeed, the rupture function is not smooth everywhere. There are kinks on the surface. We have replaced 'smooth representation' by 'smooth representation of the seabed deformation along with the deformation from its highest point towards the coast direction.'

We also add to the manuscript: "The seabed deformations change over time by multiplying an amplitude factor that increases linearly from the initial time set at 0 to the duration value t."

Lines 125-128:

A vast scientific literature exists on empirical, numerical and theoretical earthquake scaling relations, that cannot be ignored. How were these parameterisation and ranges chosen?

Turbidite event history for Cascadia subduction interface (Goldfinger et al. 2012) shows that for larger magnitudes the zone predominantly ruptures as a whole rather than concentrating high slip on smaller rupture surfaces. For the purpose of our modelling only full margin ruptures are employed and variations of the seabed deformation for such ruptures is considered. For the range of magnitudes associated with the modelled seabed deformations important for hazard assessment it is very likely that these will be full margin ruptures. This is also consistent with the view of Geological Survey of Canada who use full margin rupture for the range of magnitudes 8.4-9.2 in the recent 6[th] generation of seismic hazard model for Canada – three alternatives of rupture depth are considered having effect on the seabed deformation shape/width, not the length. Since the whole zone is rupturing we do not need to constrain further our events using relationships such as Allen and Hayes (2017) that we employed in Gopinathan (2021) where it was needed due to varying areas of ruptures.

As mentioned in a reply to a previous question, the distributions of the parameters of the seabed deformation used for generation of the predictions need to be more precisely associated with Cascadia rupture characteristics in future research, but this is out of the scope of the current paper. Nevertheless, we are able to provide a validation of the parameterisation by comparing our typical shape (mean values of the parameters for the distribution H2) against the deformation generated using the Okada solution (Fig. Y) for a dipping thrust fault with slip realistic slip distribution (Fig. X). The match is good and gives us confidence for the validity of our parameterisation. Furthermore, variations of this slip distribution result in variations of the shape of the seabed deformations as described in the paper. The last large earthquake in Cascadia happened in 1700, so detailed seabed deformation is difficult to infer, therefore random perturbations of the typical shape of the deformation are introduced. The parameter determining magnitude is mostly the maximum seabed uplift. We also provide another validation that establishes that magnitude falls into the realistic range for Cascadia subduction zone.

[Figure]

Fig. X                                                    Fig. Y

Allen TI, Hayes GP. 2017 Alternative rupture-scaling relationships for subduction interface and other offshore environments. Bulletin of the Seismological Society of America 107, 1240–1253.
Gopinathan, D., Heidarzadeh, M., & Guillas, S. (2021). Probabilistic quantification of tsunami current hazard using statistical emulation. *Proceedings of the Royal Society A*, 477(2250), 20210180.
O'Hagan, A. (2019) Expert Knowledge Elicitation: Subjective but Scientific, The American Statistician, 73:sup1, 69-81, DOI: 10.1080/00031305.2018.1518265

Line 145:

Please explain what hyperparameters are.

We added: "The hyperparameters are the parameters in the covariance function of the GP regression model. Their values are generally uncertain and can be assigned using priors or fitted to the training data by maximum likelihood estimation. As an example, a correlation length parameter is typically present, and the larger the parameter value is, the slower is the decay in the correlation between output values with respect to increasing distance from their corresponding input parameter values."

Line 146:

Please, clearly explain, at least qualitatively, the training procedure at least once throughout the paper. I understand that it is about finding outputs for unexplored inputs because of the computational cost. It must be clearly specified that as such it is a mathematical interpolation(?) based on some assumptions regarding the nature of the uncertainty, but no data were used for calibration purposes? If this is the case, this must be clearly specified as a limitation of the method quite upfront in this paper. It seems to me that quite naive initial conditions and only results of simulation (not real data) are used for calibration. Regarding the data, this is understandable, due to tsunami data scarcity. BTW, has this emulator approach ever been trained/calibrated using tsunami observations? If yes, please state it clearly, because, under a kind of ergodic hypothesis, you might assume that training elsewhere would make the usage here more trustable. Moreover, why should the emulators be independent of each other? Aren't they correlated to a degree that depends on their mutual location in the parameter space? Could you please clarify and explain better in the text? Probably, I don't understand completely because I'm not an expert on this specific

methodology but, please, make an effort to make your descriptions more accessible to the NHESS general reader.

No calibration is performed against observations due to the lack of data. We only generate future scenarios for which input ranges of values and distributions over these ranges have been determined using some past studies, current knowledge and scarce observations. To train the emulators, several different scenarios are needed which is not easy in tsunami hazard. Usually in spatial modelling one expects output values at nearby locations to be strongly correlated to each other; however, for the application at hand, some locations are close and still not strongly correlated due to variations in the local topography (e.g. a cliff, a a sandpit, a channel, etc). If we were to use a spatial correlation learned over the domain at these specific locations, we would introduce some error. We would also not gain much since our model would include another set of spatial parameters that are difficult to estimate in our context. Indeed, we have high-resolution simulations that limit the size of the number of simulation runs we can perform and therefore prohibits a reasonable estimate of the correlation structure due to the large number of output locations considered.

Line 237:

It looks like the predicted response is not defined positive as the maximum elevation. Am I wrong? Can't this be corrected?

We do not control the positivity of the predictions as represented in the histograms, however we correct it in the maps by rounding the outputs as the discrepancy is very small.

Line 238:

Underestimation (or even a bias) is generally considered worrisome in the context of the hazard assessment, particularly when conservatism is desired at a later stage by decision-makers.

Can you comment on this aspect?

We agree. We are indeed interested in the largest wave run-up in different areas, and avoiding underestimation is particularly important. However, there is not a consistent underestimation in the GP prediction. In Figure 9, the variances represent the 95% confidence intervals which means we expect that 5% of the output values are outside their corresponding confidence intervals. For example, by instead displaying intervals for the 99%-confidence level, we would expect the intervals to have been wider and possibly covered more of the outliers. For conservative risk assessments, a benefit of the GP emulation is that it provides the variance estimation so that one for instance can use the 99%-upper quartile as a conservative estimate.

Lines 271-272:

It is not clear at all what is the physical meaning of this probability. It looks like this just the probability of the impact conditional to a subjectively defined source variability modulated by a beta distribution that "can be utilised to express the scientific knowledge on the source".

What is the relation of this "probabilistic" source model to the expected future seismicity of the region? Without clarifying this, it is very difficult to say what is the relation of this "probabilistic hazard map" with the estimate of future tsunami probability, which is the core

objective of hazard assessment.

It is then hard to believe that these can be defined as "The probabilistic hazard maps in the South-East part of Vancouver island".

The statement at lines 271-272 is one of those motivating comment 3) at the beginning of this report.

Like it is, this is an interesting tool for reducing computational cost. Nothing else. And the one presented is just an illustrative example with purposely oversimplified assumptions.

It is right that this is the conditional probability, to be in agreement with the literature we modified this sentence to: "The hazard maps.."

Lines 303-304:

Here many of the concerns just expressed are even adopted by the authors. This sounds like these histograms can be translated into probabilities with some meaning.

Yet, it should be said that often also data (e.g. seismic catalogues) are used to constrain probabilities, not only expert judgement.

Thank you for this comment, we modify to: "Following similar procedures, seismic data in combination with expert knowledge on the rupture characteristics can be translated to probabilistic tsunami hazard outputs."

Line 321:

I doubt this can be judged a "real-case hazard prediction", for the reasons already expressed.

We modify: "towards realistic hazard prediction".

Line 327:

Same as above: "to produce probabilistic hazard maps that assess the tsunami potential in the area".

We modify: "The emulators' predictions were linked to their occurrence exceedance probability which allowed us to produce probabilistic hazard maps that assess the hazard intensity of such events in the area".

Lines 328-330

So, we are ready for starting coastal planning now? BTW, any comparison/validation with respect to other studies?

Or this sentences mean that a totally different approach would be needed for that? Please, clarify.

BTW, the probability of something happening in the future should be referred to a given time interval.

We modify: "This methodology could prove useful for assessing the hazard at the first stages of mitigation planning in order to take preventive measures such as built structures or natural hazard solutions."

Line 340-345:

It looks like we at least partly agree, eventually: "As so, there are some aspects that need to be considered in future work to further refine the probabilistic outputs. These span from the tsunami generation to the inundation.". This is not enough in my opinion, for the reasons that were already explained several times.

Yet, no reference to the temporal behaviour of earthquakes is made, which is necessary to address the hazard.

One last comment regards the (lack of) Discussion and the Conclusion.

The limitations of the approach are not deeply investigated, or at least their analysis not completely disclosed. First of all, it should be better clarified how much the approach may suffer from the lack of experimental data for calibration. However, this is a common issue for all the hazard assessment methods heavily based on simulations. Nevertheless, I'd like to see what would happen when trying to predict inundation and whether and to what extent nonlinearity would challenge the emulator. Moreover, no attention is paid to addressing the uncertainty related to the simulator itself - simulations are unfortunately always presented without uncertainty, we all do that - but the issue should be at least mentioned in this context. Several methods exist relying on the sensitivity of the simulators for example to the input variations. Last, how much the uncertainty on the bathymetry would affect the results?

We agree and modify: "Furthermore, gaps and mismatches in the digital elevation data should be accounted for and incorporated in the modelling for a more finely resolved representation. Uncertainties and/or errors in the bathymetry and elevation data may play a critical role in the wave elevation outputs when assessing tsunami impact at a high-resolution. Model bias is also not addressed in this study, but could be further explored in future studies. Finally, to produce a complete hazard assessment in the region, probabilistic tsunami inundation should be studied and integrated in the results. In the prediction of inundation, it is important to assess the treatment of nonlinearity by the emulators."

Technical Corrections

Figure 1:
Figure 1 is never invoked. Please check also the ordering of the other Figures, for example, Figure 8 is called out before previous ones. Check all of them.
Corrected.

Line 93:
Varying —> Various?
Corrected.

Line 153:
Add a comma "," after 2008 or remove the next one after "criterion".
Corrected.

Line 167:
Please add a link and/or reference for Gmsh.

Link was added.

Figure 5:
Where is the 0 of the profiles? Show it in the Figure or use a classic AA' BB' CC' notation for the 3 profiles and adjust the axes of the Figures to the right accordingly.
Corrected.

Line 205:
burried —> buried
Corrected.

Line 330:
"would particularly" —> "would be particularly important"? Please check this sentence.
 Corrected.

---

## Author Comment (AC2)

Referee Report #2

General comments

This study provides the application of the statistical surrogate models to predict tsunami hazards from earthquakes at CSZ. The study employs the emulation approach to quantify the tsunami hazard, particularly the maximum onshore tsunami height at Victoria, British Columbia, considering varied conditions of seabed displacement over CSZ. Overall, the formats and writing are acceptable, but I have major comments on the novelty of this work and descriptions, including the model validation, fitting process, and conclusions. Therefore, I recommend revising the manuscript and resubmit for publication.

We would like to thank the reviewer for their thorough comments and suggestions to improve the manuscript. We hope that our point-by-point answers and the modifications on the manuscript will clarify and improve the aspects of our work that need further addressing.

Specific comments
1. For the novelty of this work.
The application of the emulator for tsunami hazard modeling in CSZ tsunami, particularly in Victoria, British Columbia, is already published by the third author (Guillas et al., 2018). It is indeed the first time to check the maximum onshore tsunami height using the surrogate model, as noted by the authors. Still, the model domain seems quite similar to the previous work. The fundamentals are already provided from other references except showing the surrogate model results at a specific area in British Columbia.
I think, the authors could earn more novelty from chapter 4, which is the model validation and fitting. Still, the process is relatively limited and needed to improve to get a novelty. Here are the related comments below.

The novelty of this work expands on the methodological aspects. The use of the adaptive sequential design algorithm by Beck and Guillas (2016) is implemented for the first time to construct the GP emulators towards a realistic tsunami hazard. This was not done in Guillas et. al. (2018). This allows the use of high resolution (30 m) over a coarser one (in Guillas et al.,2018) also through the use of multi-output GPs. To improve the novelty of the manuscript, we also associate our predictions with an occurrence exceedance probability. To our best knowledge, this link is implemented for the first time. Please also see our reply to reviewer #1 (also copied below) regarding this comment:

*The novelty here is the use of the sequential design MICE by Beck and Guillas for the construction of the GP emulator of the tsunami model. This is done for the very first time towards a realistic case using High Performance Computing. Gopinathan et al (2021) and Giles et al. (2021) (now also cited in the revised manuscript) used a one shot random sampling for the training which lacks the information gain achieved by sequential design. Concretely, sequential design can reduce by 50% the computational cost, as demonstrated in (Beck, Guillas, 2016) for a set of toy problems, so applying the approach towards a realistic case is showcasing real benefits in the case of high resolution with a complex parametrization of the source and is new. Note, that Guillas et al. (2018) was neither using high resolution nor sequential design and the source was much simpler. The relationship of this work with existing literature is given below in the answers to specific comments.*

2. For model "initial validation" (Line 195).
In general, it is hard to validate numerical tsunami model results, including the generation and propagation process, due to the lack of real observed data of CSZ events. The authors conclude that the maximum water elevation at a specific point shows good agreement with Fine et al. (2018) and other references and justify the current approach. However, the current comparisons are not clear and need to be improved. To specific, the authors pick scenario 24 for the validation, which shows the best agreement(?) to others works. If

scenario 24 is chosen, it is better to show more detailed comparison results using figures or tables. The current comparison results describe the match at a particular point, but the authors need to show a spatial (map) comparison to justify the model to others. Also, it is better to provide detailed tsunami generation conditions of each reference and justify why the author chose scenario 24.

We agree. We clarify better in the manuscript:
"Scenario 24 is selected as it is the first scenario in our list of scenarios that has a maximum deformation of ca. 4m, similar to the maximum uplift in numerical studies of the event."
"The maximum uplift was used as a guideline for this comparison due to its significant contribution to the tsunami excitation. As the experimental setting is controlled by MICE, the rest of the parameter values of scenario 24 do not necessarily match with the values of other numerical studies. For example, the maximum subsidence of scenario 24 is selected to be around 1.27 m as opposed to 2 m in the buried rupture model by Fine et al. 2018. This causes some discrepancies in the wave signal, the degree of which is not calculated since the scope of this comparison is to do an initial validation of our modelling as opposed to a reproduction of the currently existing work".
We provide snapshots and spatial maps of scenario 24 but the exact locations of other works vary between them and are not always given. We added the following table of the results at the mouth of Victoria Harbour in the manuscript:

| Study | Uplift (m) | Subsidence (m) | Approximate arrival time of Wave Trough (minutes) | Approximate arrival time of Wave Crest (minutes) | Approximate wave trough (m) | Approximate wave crest (m) |
|---|---|---|---|---|---|---|
| scenario 24 | 4.09 | 1.27 | 50 | 100 | 0.2 | 1.7-1.8 |
| AECOM 2013 | 6.2 | 2.3 | Not found | 96 | 1-1.05 | 2.4-2.5 |
| Cherniwasky et al., 2007 | Not found, Mw 9 | Not found, Mw 9 | 50 | 90 | 0.5-0.6 | 1.9-2.2 |
| Fine et al 2018 | 4 | 2-2.5 | 52 | 88 | 0.96 | 1.6-1.7 |

3. For model "Fitting" (line 225)
a) As I understand, the emulator was trained from the single point in Fig. 8 (star). Need some explains why chose this point and any sensitivities on training and results.
The point in Fig. 8 (now Fig. 4 of the document) was selected to drive the design algorithm for the selection of the experiments. We add in the text: "This design location was selected as it provides variance in the response driven by each scenario and it is close to the centre of the region of impact. As it directs the sequential design there is some sensitivity of the design to this point, but in our opinion not that large as another point in the region would yield similar results since the influence of the parameters on impact points varies but not much. Furthermore, small variations in the design of experiments obtained have little influence on the construction of the emulator, but an agnostic one-shot design would greatly

differ from any of the sequential designs obtained by our approach and be much less efficient as it would ignore completely the concrete influence of the inputs on outputs to efficiently design the computer experiment. "

b) Figure 9 shows the performance of the prediction from the emulator at three different points as the authors noted that the surrogate model couldn't capture the pattern well at location 2 but show good agreement at location 3. Can authors explain why they are so different?

We add:
"As the waves propagate on land, the prediction becomes more challenging due to even the slightest variations caused by the surrounding topography. The sensitivity of the locations to the variance in the scenarios also plays a significant role. Location 2, for example, does not show large sensitivity to the variation of the parameters, the maximum elevation is close to zero in all of the cases.  Location 3 is closer to the source and is experiencing the highest wave run-up and it is, therefore, less affected by these slight variations in the topography but more sensitive to the varying scenarios."

c) Due to the small number of comparisons and fluctuation, those three points are not enough to represent the overall performance of the emulator. It is questionable that location 001 (I think it is location 01) could represent the general pattern of RMSE error of the emulator index (Fig. 9d). It is hard to conclude that 50 and 60 are good enough for your hazard results in Fig. 12. It is recommended to show more comparison results (somewhat similar to Fig. 9a,b,c but more efficiently) at different locations. We can observe a spatial variation of wave height at the shoreline in Fig. 12. The authors may consider checking the variance of RMSE for the emulator index at different locations.

Indeed, Location 1 is not in a general sense representing the RMSE error at all locations. The 3 locations are used for illustrative purposes, in order to demonstrate the emulators' behavior at some points. Showing more locations might not necessarily add to the manuscript's content as they may exhibit similar behavior, however the relative error might increase at locations further inland. This is related to the previous comment, and we also add:

"The RMSE is computed at these three locations for illustrating the behavior of the emulator's predictions at certain points, the relative error might increase further inland at inundated locations."

and also: "GP emulation is well suited for approximating nonlinear simulation behaviors, even when considering continuous outputs of low regularity and when restricted to small-sized experimental designs with space-filling properties. As shown by Owen et al. (2017), when considering two cases with computationally intensive simulators, more specifically, a land-surface simulator and a launch vehicle controller, GP emulation demonstrates good approximation properties even for small design sizes. By small design sizes, we refer to designs with the number of samples being about ten times the number of input parameters, a widely used rule of thumb for effective computer experiment design (Loeppky et al., 2009)."

Owen et al. (2017) "Comparison of surrogate-based uncertainty quantification methods for computationally expensive simulators. SIAM/ASA Journal on Uncertainty Quantification, 5(1): 403-435.

Loeppky et al. (2009) "Choosing the sample size of a computer experiment: A practical guide", Technometrics, 51: 366-376.

4. About Probabilistic tsunami hazard,
One of the conclusions (at line 326) is that the emulator allows probabilistic hazard maps.
Fig. 11 and 12 are not a typical probabilistic hazard map, which is not commonly accepted.
The probabilistic tsunami hazard map provides the map (spatial distributions) of tsunami
intensity measures (e.g., maximum tsunami flow depth) at a specific recurrence interval such
as 500 yr, 1,000 yr, and 2,500 yr through a probabilistic tsunami hazard analysis (PTHA).
The authors need to clarify that the current work from the typical probabilistic hazard map.
Also, it is hard to agree that the current work is a kind of alternative format of hazard map as
described in line 328. It is misleading to readers.
We agree that this is an important issue that needs to be addressed. Please see our
response to the reviewer #1 below in an attempt to address this issue.

*We agree and we refrained from using the explicit term PTHA for our work. The emulators
can give the probability of a tsunami intensity measure to exceed a certain threshold given
predefined metrics at any horizon. We now provide a correspondence between predictions
and time horizons. We make a first attempt here, which is simplistic but we believe adds to
the novelty of the manuscript as it is, to our knowledge, the first to incorporate emulators for
probabilistic tsunami hazard (as the term is currently accepted). The following section will be
added and will be discussed in more detail in the revised manuscript:*

*"Further, we associate the scenarios and predictions of the H2 distribution with probability of
exceedance. We study the pattern of 1/1500 year exceedance rate wave heights over the
region drawing elements from the process followed in Park et al. (2017). The probability of a
full-rupture generated tsunami is considered, the earthquake magnitudes associated with
such an event are in the ranges of Mw=8.7-9.3 (Goldfinger et al., 2012). We link the moment
magnitude for each event with the maximum seabed uplift caused by the linearly decaying
slip on the rupture surface similar to the Cascadia subduction interface, a simple linear
solution, first introduced by Okada (1985) is used. Following this approach, the magnitudes
of the H2 scenarios range between 8.77-9.28. To associate frequency of events with
earthquake magnitudes, a Tapered Gutenberg–Richter (TGR) distribution is utilized which
has been proved to give adequate predictions for the Cascadia subduction zone (Rong et
al., 2014). The TGR Complementary Cumulative Distribution function for a given earthquake
magnitude $m$ can be computed as:*

$$F(m) = \left[10^{1.5(m_t - m)}\right]^{\beta} exp\left[10^{1.5(m_t - m_c)} - 10^{1.5(m - m_c)}\right],$$

*where $m_t$ is a threshold magnitude above which the catalogue is assumed to be complete
(here $m_t = 6.0$), $m_c$ is the corner magnitude and $\beta$ is the index parameter of the distribution.
Considering the 10,000-year paleoseismic record, as reconstructed by Goldfinger et al.
(2012) from turbidite data, $m_c$ and $\beta$ take values of 9.02 and 0.59 respectively. The discrete
number of $m_j$ magnitudes can also be computed by:*

$$P\left[M = m_j\right] = G\left(m_j + 0.5\Delta m\right) - G(m_j - 0.5\Delta m),$$

*where $G(m) = 1 - F(m)$ denotes the cumulative density function and $\Delta m$ is the discretization
interval. Following the above, the mean annual rate of exceedance and the number of
earthquakes in 200K years can be estimated as shown in Figure 13a. These relationships
can then be used to associate the probability of exceedance for each event of the event set
H2 by computing the relative frequency of each event within the data (Fig.13b). The
occurrence exceedance probability for the hazard values $h_1 > h_2 > \cdots > h_n$, with $n = 2000$
becomes $P_{ex}(h_1) = 0$ for $h_1$ (corresponding magnitude is equal to 9.3), $P_{ex}(h_2) = f_1$, and for
the other hazard values can be computed as:*

$$P_{ex}(h_{i+1}) = 1 - (1 - P_{ex}(h_i))(1 - f_i) \quad for \quad i = 2,3 \dots, n,$$

*where $f_i$ is the event frequency assigned according to the magnitude bin to which the event was linked by its maximum uplift. The above relation comes from basic probability theory and is often used e.g. in Monte Carlo probabilistic seismic hazard assessment (Musson, 2000). Accordingly, the mean annual exceedance rate can be computed for the hazard values at each location (Fig. 13c). Considering then a 1/1500 exceedance rate, the hazard curves of each location can be used to construct the hazard maps of Figure 13d which represent the maximum wave heights from the H2 events.*

[Figure]

*Figure 13*

*We note that as a first demonstration of how the emulators' predictions can be linked with the probability of exceeding a tsunami intensity measure over time, this is a simplified case. To better capture the probabilistic tsunami hazard in the region the seabed deformation parameter distributions used for generation of the predictions need to be precisely associated with Cascadia rupture characteristics, other more realistic magnitude-frequency relationships than the selected Tapered Gutenberg–Richter distribution can be considered in future research."*

No technical comments.

---

## Author Response (AR2)

Referee Report

Review comments

The reasons for choosing three design locations are not sufficiently clarified in the revised version too. Why are these three points good enough? Authors responsed that there will be small variances, but it would be better to quantify this by comparing the results by adding additional points in the sequential design.

Also, it is still not clear that how these three points are representative locations for sequential design. How do these points represent all different bathymetry conditions in the test area? What are the depth and slope condition of three locations? The reason that it is near the center region is not enough. The results show that the Location 3 provided much larger Hmax than Location1 and 2, but the results of Location 1 and 2 are too small.

We would like to thank the reviewer for their comments which help us to clarify the content of the manuscript. The sequential design approach uses one location to decide on the design, not three. The three locations are for illustration of the predictions to display variations in the outputs. We apologize for the lack of clarity and improved the paper accordingly in the text.

The referee is right that only one location chosen is driving the design. Our argument is that it is much better than zero, (which is often the case in conventional space filling designs, like Latin hypercube sampling), as justified in previous published studies. Evidently adding more locations could improve the design further, but some methodological statistical developments should be first established to decide on a strategy to actually benefit from using more points. Indeed, the design improvements are unlikely to be improved much with the addition of more points since the characteristics of the earthquake source are influencing all locations in a similar way as the parameterization is geographically wide.  We improved the text accordingly.

We picked these 3 locations with depth nearly zero: one close to the design (#1), one further away from the design point (#2) but with elevation similar to #1, and #3 close to #1 but with a very different elevation. The referee is right that these outputs are different since we wanted to show some diversity and indeed are representative. More points would help show more variations, but would lengthen unnecessarily the paper. We explained and improved the text accordingly.